# Graph Robustness Benchmark: Benchmarking the Adversarial Robustness of Graph Machine Learning

**Qinkai Zheng**[†], **Xu Zou**[†], **Yuxiao Dong**[†], **Yukuo Cen**[†],
**Da Yin**[†], **Jiarong Xu**[∘], **Yang Yang**[⋄], **Jie Tang**[†*]

[†] Department of Computer Science and Technology, Tsinghua University
[∘] Fudan University [⋄] Zhejiang University
{qinkai, yuxiaod, jietang}@tsinghua.edu.cn
{zoux18, cyk20, yd18}@mails.tsinghua.edu.cn
jiarongxu@fudan.edu.cn, yangya@zju.edu.cn

## Abstract

Adversarial attacks on graphs have posed a major threat to the robustness of graph machine learning (GML) models. Naturally, there is an ever-escalating arms race between attackers and defenders. However, the strategies behind both sides are often not fairly compared under the same and realistic conditions. To bridge this gap, we present the Graph Robustness Benchmark (GRB) with the goal of providing a *scalable*, *unified*, *modular*, and *reproducible* evaluation for the adversarial robustness of GML models. GRB standardizes the process of attacks and defenses by 1) developing scalable and diverse datasets, 2) modularizing the attack and defense implementations, and 3) unifying the evaluation protocol in refined scenarios. By leveraging the GRB pipeline, the end-users can focus on the development of robust GML models with automated data processing and experimental evaluations. To support open and reproducible research on graph adversarial learning, GRB also hosts public leaderboards across different scenarios. As a starting point, we conduct extensive experiments to benchmark baseline techniques. GRB is open-source and welcomes contributions from the community. Datasets, codes, leaderboards are available at https://cogdl.ai/grb/home.

## 1  Introduction

Graph machine learning (GML) models, from network embedding [1, 2, 3] to graph neural networks (GNNs) [4, 5, 6, 7, 8, 9], have shown promising performance in various domains, such as social network analysis [1], molecular graphs [5], and recommender systems [10]. However, GML models are known to be vulnerable to adversarial attacks [11, 12, 13, 14, 15, 16, 17, 18]. Attackers can modify the original graph by adding or removing edges [11, 19, 20], perturbing node attributes [12, 13, 14, 15], or injecting malicious nodes [16, 17, 18] to conduct adversarial attacks. Despite the relatively minor changes to the graph, the performance of GML models can be impacted dramatically.

Threatened by adversarial attacks, a line of attempts have been made to have robust GML models. For example, recent GNN architectures such as RobustGCN [21], GRAND [22], and ProGNN [23] are designed to improve the adversarial robustness of GNNs. In addition, pre-processing based methods, such as GNN-SVD [24] and GNNGuard [25], alleviate the impact of attacks by leveraging the intrinsic graph properties and thus improve the model robustness. Despite various efforts in this direction, there are several common limitations from both the attacker and the defender sides:

---

[*]Jie Tang is the corresponding author.

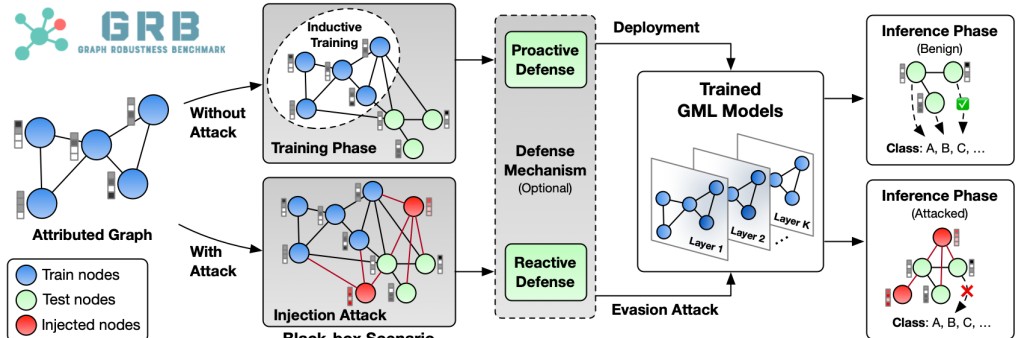

Figure 1: An example of GRB's attack vs. defense (*graph injection*) scenario: *Black-box*: attackers only have access to the attributed graph but not the target models; *Inductive*: target models are trained in an inductive setting (test nodes are unseen during training); *Injection*: attackers are allowed to inject new nodes without modifying the existing ones; *Evasion*: attacks happen during model inference. All attacks and defenses are evaluated under unified settings to be fairly compared.

1. **Unrealistic Attack/Defense Scenarios.** The existing attack and defense setups are often ambiguously defined with unrealistic assumptions, such as ignoring the real-world capabilities of attackers and defenders, resulting in less practical applications.

2. **Lack of A Unified Evaluation Protocol.** Previous works often use different settings (*e.g.*, datasets, data splittings, attack constraints) in their experiments, resulting in biases in the evaluation and thus making it difficult to fairly compare different methods.

3. **Lack of Scalability.** Most existing attacks and defenses are performed on very small-scale graphs (*e.g.*, <10,000 nodes) without considering different levels of attack/defense difficulties, which are far from the scale and complexity of real-world applications.

To date, there exist several well-established GML benchmarks. For example, the Open Graph Benchmark (OGB) [26] offers abundant datasets and a unified evaluation pipeline for GML. Benchmarking GNNs [27] is a standardized benchmark with consistent experimental settings. However, they mainly focus on evaluating the performance of GML models, regardless of their robustness. DeepRobust [28] is a toolkit with implementations of attacks and defenses on both image and graph data, which by design is not a GML benchmark. Therefore, to address the aforementioned limitations, there is an urgent need for public benchmarks on evaluating the *adversarial robustness* of GML models.

In this paper, we propose the Graph Robustness Benchmark (GRB)—the first attempt to benchmark the adversarial robustness of GML models. The goal of GRB is to provide a reproducible framework that enables a fair evaluation for both adversarial attacks & defenses on GML models under unified settings. To achieve this, GRB is designed to have the following properties:

1. **Refined Attack/Defense Scenarios.** GRB includes two refined attack scenarios: *graph modification* and *graph injection*, covering the majority of works in the field. By revisiting the limitations of previous works, we formalize precise definitions for both attackers' and defenders' capabilities, including available information to use and allowed actions, forming more realistic evaluations.

2. **Scalable and Unified Evaluations.** GRB contains various datasets of different orders of magnitude in size, with a specific robustness-focused splitting scheme for various levels of attacking/defending difficulties. It also provides a unified evaluation pipeline that calibrates all experimental settings, enabling fair comparisons for both attacks and defenses.

3. **Reproducible and Public Leaderboards.** GRB offers a modular code framework* that supports the implementations of a diverse set of baseline methods covering GML models, attacks, and defenses. Additionally, it hosts public leaderboards across all evaluation scenarios, which will be continuously updated to track the progress in this community.

Overall, GRB serves as a *scalable*, *unified*, *modular*, and *reproducible* benchmark on evaluating the adversarial robustness of GML models. It is designed to facilitate the robust developments of graph adversarial learning, summarizing existing progress, and generating insights into future research.

---

*https://github.com/THUDM/grb

## 2 Adversarial Robustness in Graph Machine Learning

### 2.1 Problem Definition

In graph machine learning, adversarial robustness refers to the ability of GML models to maintain their performance under potential adversarial attacks. Take the task of node classification as an instance, for an undirected attributed graph $\mathcal{G} = (\mathcal{A}, \mathcal{F})$ where $\mathcal{A} \in \mathbb{R}^{N \times N}$ represents the adjacency matrix of $N$ nodes and $\mathcal{F} \in \mathbb{R}^{N \times D}$ denotes the set of node features with $D$ dimensions. Define a GML model $\mathcal{M} : \mathcal{G} \to \mathcal{Z}$ where $\mathcal{Z} \in [0, 1]^{N \times L}$, which maps a graph $\mathcal{G}$ to probability vectors with $L$ classes. Generally, the objective of adversarial attacks on GML models can be formulated as:

$$\max_{\mathcal{G}'} |\arg\max_{l \in [1,...,L]} \mathcal{M}(\mathcal{G}') \neq \arg\max_{l \in [1,...,L]} \mathcal{M}(\mathcal{G})| \text{ s.t. } d_{\mathcal{A}}(\mathcal{A}', \mathcal{A}) \leq \Delta_{\mathcal{A}} \text{ and } d_{\mathcal{F}}(\mathcal{F}', \mathcal{F}) \leq \Delta_{\mathcal{F}} \quad (1)$$

where $\mathcal{G}' = (\mathcal{A}', \mathcal{F}')$ is the attacked graph, and $d_{\mathcal{A}}$ and $d_{\mathcal{F}}$ are distance metrics in the metric space $(\mathcal{A}, d_{\mathcal{A}})$ and $(\mathcal{F}, d_{\mathcal{F}})$. The attacker tries to maximize the number of incorrect predictions by GML models, under the constraints $\Delta_{\mathcal{A}}$ and $\Delta_{\mathcal{F}}$. For instance, $\Delta_{\mathcal{A}}$ can be the limited number of modified edges and $\Delta_{\mathcal{F}}$ can be the limited range of modified features (*Cf.* Section 3 for detailed discussions).

### 2.2 Revisiting Adversarial Attacks and Defenses in GML

In the work of Szegedy *et al.* [32], the existence of adversarial examples was revealed for ML models in image classification—imperceptible perturbations on inputs have ineligible impact on outputs of models. Recent works (in Table 1) show that GML models are no exception. Graph adversarial attacks can mainly be categorized into two types according to the attack approach: *graph modification* attack and *graph injection* attack. Graph modification attacks directly modify the existing graph, by adding or removing edges (*e.g.*, DICE [19], FGA [11], FLIP [29], NEA [29], STACK [31]), or further modifying node features (*e.g.*, Nettack [12], FGSM [12], RL-S2V [30], Metattack [13]). Differently, graph injection attacks add new malicious nodes without modifying the original graph (*e.g.*, AFGSM [16], SPEIT [17], TD-GIA [18]). Facing the problem of scalability, some attacks are not applicable to large graphs due to their high time complexity [12, 13, 30] or expensive memory consumption [11, 29].

Table 1: A categorization of graph adversarial attacks. There are mainly two scenarios: *graph modification* and *graph injection*. GRB supports the implementation of all types of methods. [†]

| Adversarial Attack | Knowledge | | Objective | | Approach | | Scalability |
|---|---|---|---|---|---|---|---|
| | *Black.* | *White.* | *Poi.* | *Eva.* | *Mod.* | *Inj.* | |
| DICE [19] | ✔ | – | ✔ | – | ✔ | – | ✔ |
| FGA [11] | ✔ | – | ✔ | – | ✔ | – | ✗ |
| FLIP [29] | ✔ | – | ✔ | – | ✔ | – | ✔ |
| NEA [29] | ✔ | – | ✔ | – | ✔ | – | ✗ |
| FGSM [12] | ✔ | ✔ | ✔ | – | ✔ | – | ✔ |
| Nettack [12] | ✔ | ✔ | ✔ | – | ✔ | – | ✗ |
| RL-S2V [30] | ✔ | ✔ | ✔ | – | ✔ | – | ✗ |
| Metattack [13] | ✔ | – | ✔ | – | ✔ | – | ✗ |
| STACK [31] | ✔ | – | ✔ | – | ✔ | – | ✗ |
| AFGSM [16] | ✔ | – | ✔ | – | – | ✔ | ✔ |
| SPEIT [17] | ✔ | – | – | ✔ | – | ✔ | ✔ |
| TDGIA [18] | ✔ | – | – | ✔ | – | ✔ | ✔ |
| GRB Mod. Scenario | ✔ | – | – | ✔ | ✔ | – | ✔ |
| GRB Inj. Scenario | ✔ | – | – | ✔ | – | ✔ | ✔ |
| GRB Support | ✔ | ✔ | ✔ | ✔ | ✔ | ✔ | ✔ |

[†] The table represents the original settings, while methods can be adapted to other settings by using GRB's modualr coding framework.

Defenses can mainly be categorized into two types: *preprocess-based* defense and *model-based* defense. The first type regards the attacked graphs as noisy ones and defenders can preprocess the adjacency matrix (*e.g.*, GNN-SVD [24], GNN-Jaccard [33]) or the features of nodes (*e.g.*, feature transformation [17]), to alleviate the effect of perturbations. The second type achieves robustness through *model enhancement*, either by robust training schemes (*e.g.*, adversarial training [34, 35]) or new model architectures (*e.g.*, RobustGCN [21], GNNGuard [25]). Some defenses also suffer from the problem of scalability, due to the need of calculation on large dense matrices [24, 33, 25].

Notwithstanding the significant progress, existing works share some common limitations: (1) Lack of scalability: Most works only consider very small graphs and cannot be scaled up to larger ones due to time/memory complexity. (2) Lack of generalization: Most attacks/defenses are evaluated on very basic GML models, but not on other variants. Meanwhile, some methods are only effective for specific models with ad-hoc designs, which makes the results less generalized and practical. (3) Ill-defined scenarios: The scenarios and assumptions proposed in some previous works are not realistic, *e.g.*, the *unnoticeability* under *poisoning* setting ignores the real capability of the defenders (*Cf.* Appendix A.3 for details). Besides, there are no unified standards on evaluating the adversarial robustness. Different settings (*e.g.*, the choice of datasets, random splitting, different constraints) introduce biases, which makes it hard to compare the effectiveness of different methods. In light of these challenges, there is an urgent need for benchmarking the adversarial robustness of GML.

# 3 GRB: Graph Robustness Benchmark

## 3.1 Overview of GRB

To overcome the limitations of previous works, we propose the Graph Robustness Benchmark (GRB)—a standardized benchmark for evaluating the adversarial robustness of GML. To ensure GRB's scalability, we include datasets of different sizes with scalable attack/defense baselines. To have a unified process, we standardize the evaluation scenarios with precise constraints and realistic assumptions on attackers and defenders. To make GRB easy-to-use, we provide a modular pipeline that facilitates the implementation of GML models, attacks, and defenses. To guarantee the reproducibility, we open-source and maintain the GRB public leaderboards that are continuously updated to track the progress of the community.

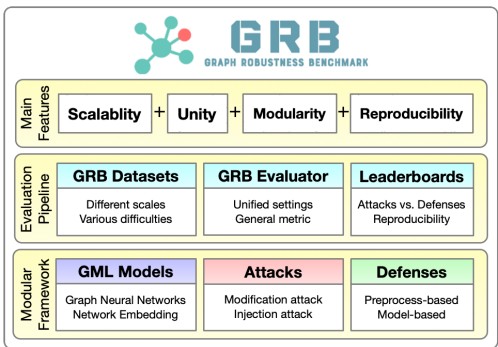

Figure 2: GRB Framework.

Altogether, GRB serves as a *scalable*, *unified*, *modular*, *reproducible* benchmark on evaluating the adversarial robustness of GML models. We present the solutions to achieve these goals for GRB.

## 3.2 The Unified Evaluation Scenario of GML Adversarial Robustness

To evaluate the adversarial robustness, it is essential to be aware of the capabilities of potential attackers. We categorize attacks into the following aspects (as shown in Table 1):

1. **Knowledge.** *Black-box*: Attackers do NOT have access to the targeted model (including its architecture, parameters, defense mechanism, etc.). However, they can access the graph data (structure, features, labels of training data, etc.). Additionally, they have limited chances to query the model to get outputs. *White-box*: Attackers have access to ALL information. However, if the targeted model has a random process, the run-time randomness is still preserved.

2. **Objective.** *Poisoning*: Attackers generate corrupted graph data and assume that the targeted model is (re)trained on these data to get a worse model. *Evasion*: The target model has already been trained, and attackers can generate corrupted graph data to affect its inference.

3. **Approach.** *Modification*: Attackers modify the original graph (the same one used by defenders for training) by adding/removing edges or perturbing node features. *Injection*: Attackers do not modify the original graph but inject new malicious nodes to influence a set of targeted nodes.

In practice, the most common real-world case is that the GML models have already been trained for specific tasks and deployed in a secret way, i.e., *black-box* and *evasion* settings. Thus, in GRB, we propose two unified evaluation scenarios under these settings, *graph modification* and *graph injection*.

**Graph Modification.** This has been the most studied scenario, in which attackers can directly modify the graph (by adding/removing edges or perturbing node attributes) to attack the GML models. Under real-world conditions, this is theoretically possible but practically difficult, as the modification attacks require the authority to access the target nodes in order to to change their contents. Nevertheless, this scenario enables us to understand how the GML models behave under intended modifications.

**Graph Injection.** This scenario was first introduced in the KDDCUP 2020 task of Graph Adversarial Attacks & Defenses[†], which targeted at injecting new nodes to a large-scale academic graph. It is more realistic than the modification one since injecting new nodes is more practically possible than modifying the existing ones. However, the task in KDDCUP 2020 considers a transductive setting, *i.e.*, test nodes (except for their labels) are available during training. In this case, defenders can simply memorize benign nodes and identify the injected nodes, making it an imperfect setting.

Thus, to further GRB's practical usage (*Cf.* Appendix A.3 for detailed discussions), we make the following assumptions for both scenarios: (1) Black-box: Both attackers and defenders do *not* have

---

[†]https://www.biendata.xyz/competition/kddcup_2020_formal/

knowledge about the methods each other applied. (2) Inductive: The GML models are trained in trusted data and used to classify unseen data (*e.g.*, new users), *i.e.*, the validation and test data is unseen during training. (3) Evasion: Attacks will only happen during the inference phase. Furthermore, we clarify attackers' and defenders' capabilities in GRB:

1. For attackers: (a) They have knowledge about the entire graph (including all nodes, edges and labels but excluding the labels of the test nodes), but do not have knowledge about the target model or defense mechanism. (b) For graph modification, following the most common setting in previous works, attackers are allowed to perturb a limited number of edges in the graph ($\Delta_{\mathcal{A}}$: the number of modified edges less than a ratio $\gamma_e$ of all edges). (c) For graph injection, we follow the heuristic setting of KDDCUP 2020, attackers are allowed to inject new nodes with limited edges ($\Delta_{\mathcal{A}}$: less than $N_n$ injected nodes each with less than $N_e$ edges; $\Delta_{\mathcal{F}}$: constrained range of features $[\mathcal{F}_{min}, \mathcal{F}_{max}]$.). (d) They are not allowed to modify the original graph for training. (e) They are allowed to get predictions from the target model through a limited number of queries.

2. For defenders: (a) They have knowledge about the graph excluding the test nodes to be attacked. (b) They are allowed to use any method to increase the adversarial robustness, but do not have prior knowledge about the edges/nodes that are modified/injected.

3. For both sides: Attackers/defenders can of course make assumptions even in the black-box scenario. For instance, attackers can assume that the target system deploys a certain type of GML models, then it can be used as the surrogate model to conduct transfer attacks. Moreover, it is not reasonable to assume that the defense mechanism can be completely held secretly, known as the Kerckhoffs' principle [36]. If a defense wants to be general and universal, it should guarantee part of the robustness even when attackers have some knowledge about it. In GRB, we evaluate an attack vs. multiple defenses (vice versa), thus the assumptions can hardly violate the *black-box* conditions. As a result, the objective for both sides is to be generally effective against all potential methods rather than just a single one.

By following the above rules, we provide unified evaluation scenarios for attacks and defenses in a principled way. It is worth noting that these unified scenarios are not the only valid ones, GRB will include more scenarios as this field eveloves over time.

### 3.3 The Modular GRB Pipeline

GRB offers a modular pipeline, which is based on PyTorch [37] as well as other popular GML libraries like CogDL [38] and DGL [39]. Specifically, it contains the following modules: (1) Dataset: GRB provides data-loaders for GRB datasets and applies necessary preprocessing including splitting and feature normalization; it also supports external datasets like OGB [26] or user-defined datasets. (2) Model: The GML models are implemented based on PyTorch, CogDL, and DGL and GRB can automatically transform inputs to compatible formats. (3) Attack: We implement adversarial attacks by abstracting the attack process to different components, e.g., graph injection attacks are decomposed to node injection and feature generation. (4) Defense: GRB engages defense mechanisms to GML models, including *preprocess-based* and *model-based* ones. (5) Evaluator: The attack or defense methods are evaluated under unified settings and metrics. Essentially, GRB unifies and modularizes the entire process, including loading datasets, training/loading models, applying attacks/defenses, and generating the evaluation results; it also helps to reproduce the exact results on GRB leaderboards. In addition to these modules, GRB also offers other functions including *Trainer* for model training, *AutoML* for automatic parameter search, and *Visualise* for visualizing the attack process.

The GRB framework has the following features: (1) Easy-to-use: the baseline methods are easy to use by only a few lines of codes, as shown in Figure 3. (2) Fair-to-compare: all methods are fairly compared under unified settings. (3) Up-to-date: the leaderboards for each dataset are maintained to continuously track the progress in the domain. (4) Reproducible: GRB prioritizes reproducibility. All necessary materials are made public to reproduce results on leaderboards, including the trained models, generated attack results, etc. Users can reproduce results by a single command line (*Cf.* Appendix A.5 for GRB reproducibility rules). All codes are available in https://github.com/THUDM/grb, where the implementation details and examples can be also found. The API documentations are covered in https://grb.readthedocs.io/en/latest/.

```
import torch # pytorch backend                  from grb.attack.tdgia import TDGIA
from grb.dataset import Dataset
from grb.model.torch import GCN                  # Attack configuration
from grb.utils.trainer import Trainer            tdgia = TDGIA(lr=0.01,
                                                               n_epoch=10,
# Load data                                                   n_inject_max=20,
dataset = Dataset(name='grb-cora', mode='easy',               n_edge_max=20,
                  feat_norm='arctan')                         feat_lim_min=-0.9,
# Build model                                                 feat_lim_max=0.9,
model = GCN(in_features=dataset.num_features,                 sequential_step=0.2)
            out_features=dataset.num_classes,    # Apply attack
            hidden_features=[64, 64])            rst = tdgia.attack(model=model,
# Training                                                          adj=dataset.adj,
adam = torch.optim.Adam(model.parameters(), lr=0.01)               features=dataset.features,
trainer = Trainer(dataset=dataset, optimizer=adam,                 target_mask=dataset.test_mask)
                  loss=torch.nn.functional.nll_loss)  # Get modified adj and features
trainer.train(model=model, n_epoch=200, dropout=0.5, adj_attack, features_attack = rst
              train_mode='inductive')
```

Figure 3: GRB usage examples. Left: Train GCNs on the *grb-cora* dataset. Right: Apply the TDGIA attack on the trained model. GRB facilitates the usage of GML models, attacks,and defenses.

## 3.4 The GRB Baselines

Currently, GRB covers a rich set of baselines for the GML models, attacks, and defenses.

**Seven GML models**: GCN [4], GAT [6], GIN [7], APPNP [8], TAGCN [20], GraphSAGE [5], SGCN [9]. Note that these models are not originally designed to increase robustness.

**Twelve Attacks**: Seven modification attacks—RND [12], DICE [19], FGA [11], FLIP [29], NEA [29], STACK [31], and PGD [34]—and five injection attacks—RND, FGSM [40], PGD [34], SPEIT [17], and TDGIA [18]. More details can be found in Appendix A.4.2.

**Five Defenses**: GRB adopts RobustGCN (R-GCN) [21], GNN-SVD [24], and GNNGuard [25]. Additionally, we find that techniques like layer normalization (LN) [41] and adversarial training (AT) [34], if properly used in the proposed evaluation scenarios, can significantly increase the robustness of various GML models. The LN can be applied on the input features and after each graph convolutional layer (except for the last one). The idea is to stabilize the dynamics of input and hidden states to alleviate the impact of adversarial perturbations. The AT uses modification/injection attacks during training to make GML models more robust. Note that most of previous works only use AT to perturb the existing graph, however, we find that AT also works well by injecting new nodes during training. These two defenses are general and scalable, and the experiment results show that they outperform previous dedicated methods. Thus, we include them in GRB as strong baselines for defenses. More details can be found in Appendix A.4.3.

## 3.5 The GRB Datasets

Table 2: Statistics of five GRB datasets covering from small- to large-scale graphs.

| Dataset | Scale | #Nodes | #Edges | #Feat. | #Classes | Feat. Range (original) | Feat. Range (normalized) |
|---------|-------|--------|--------|--------|----------|------------------------|--------------------------|
| *grb-cora* | Small | 2,680 | 5,148 | 302 | 7 | [-2.30, 2.40] | [-0.94, 0.94] |
| *grb-citeseer* | Small | 3,191 | 4,172 | 768 | 6 | [-4.55, 1.67] | [-0.96, 0.89] |
| *grb-flickr* | Medium | 89,250 | 449,878 | 500 | 7 | [-0.90, 269.96] | [-0.47, 1.00] |
| *grb-reddit* | Large | 232,965 | 11,606,919 | 602 | 41 | [-28.19, 120.96] | [-0.98, 0.99] |
| *grb-aminer* | Large | 659,574 | 2,878,577 | 100 | 18 | [-1.74, 1.62] | [-0.93, 0.93] |

**Scalability.** GRB includes five datasets of different scales, *grb-cora*, *grb-citeseer*, *grb-flickr*, *grb-reddit*, and *grb-aminer*. The original datasets are gathered from previous works [42, 43, 18] and are reprocessed for GRB. The basic statistics of these datasets are shown in Table 2. More details about datasets can be found in Appendix A.1.

**Data Splitting.** GRB introduces a new splitting data scheme designed for evaluating the GML adversarial robustness. Its key idea is based on the assumption that nodes with lower degrees are easier to attack, as demonstrated in [18]. If a target node has few neighbors, it is more likely to be influenced by adversarial perturbations aggregated from its neighbors. Thus, we construct test subsets

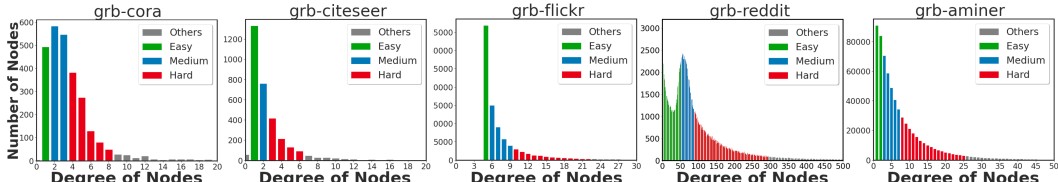

Figure 4: GRB's splitting scheme. Difficulties are related to the average degree of test nodes.

with different average degrees to represent different difficulties. First, we rank all nodes by their degrees. Second, we filter out 5% nodes with the lowest degrees (*e.g.*, isolated nodes that are too easy to attack) and 5% nodes with the highest degrees (*e.g.*, nodes connected to hundreds of other nodes that are too hard to attack). Third, we divide the rest of nodes into three equal partitions without overlapping, and randomly sample 10% nodes (without repetition) from each partition. Finally, we get three test subsets with different degree distributions as shown in Figure 4, which are defined as Easy/Medium/Hard/Full ('E/M/H/F') with 'F' containing all test nodes. For the rest of nodes, we divide them into the training set (60%) and validation set (10%).

**Feature Normalization.** Initially, the features in each dataset have various ranges. To unify their constraints and to have values in the same scale (*e.g.*, range $[-1, 1]$), we apply a standardization followed by an arctan transformation: $\mathcal{F} = \frac{2}{\pi} \arctan(\frac{\mathcal{F} - \text{mean}(\mathcal{F})}{\text{std}(\mathcal{F})})$. The statistics of datasets after the splitting scheme and the feature normalization can be found in Appendix A.1.

## 4 Experiments

With the support of GRB's modular framework, we conduct extensively experiments to evaluate the adversarial robustness of GML models under the unified evaluation protocol, from which insights are generated into the developments of the field.

### 4.1 Experimental Settings

**Baselines.** (1) For GML models, we include 7 baselines: GCN [4], GAT [6], GIN [7], APPNP [8], TAGCN [20], GraphSAGE [5], SGCN [9]. All models are salable to large graphs. (2) For modification attacks, we include 7 baselines: RND, DICE [19], FGA [11], FLIP [29], NEA [29], STACK [31], and PGD [34], among which RND, DICE, FLIP, and PGD are scalable to large graphs. FGA, NEA, and PGD need to train a surrogate model to conduct transfer attacks. (3) For injection attacks, we include 5 baselines: RND, FGSM [40], PGD [34], SPEIT [17], TDGIA [18]. They are all scalable and FGSM, PGD, SPEIT, TDGIA need to train a surrogate model to conduct transfer attacks. (4) For defenses, we include R-GCN [21], GNN-SVD [24], GNNGuard [25]. Among which only R-GCN is scalable, since the other two methods require calculation on dense adjacency matrix. Thus, we also adapt two general defense methods, layer normalization (LN) [41] and adversarial training (AT) [34] to the proposed scenarios. More details and hyper-parameter settings can be found in Appendix A.4 A.5.

**Evaluation Metrics.** For attacks: (1) Avg.: Average accuracy of all defenses (including vanilla GML models). (2) Avg. 3-Max: Average accuracy for the 3 most robust methods. (3) Weighted: Weighted accuracy, calculated by: $s_w^{\text{ATK}} = \sum_{i=1}^{n} w_i s_i, w_i = \frac{1/i^2}{\sum_{j=1}^{n}(1/j^2)}, s_i = (S_{descend}^{\text{DEF}})_i$ where $S_{descend}^{\text{DEF}}$ is the set of defense scores in a descending order. The metric attaches more weight to more robust methods. For defenses: (1) Avg.: Average accuracy of all attacks. (2) Avg. 3-Min: Average accuracy of the 3 most effective attacks. (3) Weighted: Weighted accuracy across various attacks, calculated by: $s_w^{\text{DEF}} = \sum_{i=1}^{n} w_i s_i, w_i = \frac{1/i^2}{\sum_{j=1}^{n}(1/j^2)}, s_i = (S_{ascend}^{\text{ATK}})_i$ where $S_{ascend}^{\text{ATK}}$ is the set of attack scores in an ascending order. The metric attaches more weight to more effective attacks.

### 4.2 Experimental Results

We show an example of GRB leaderboard, robust ranking of GML models, and various factors that affect the adversarial robustness in GML. More results can be found in Appendix and on our website.

**An Example of the GRB Leaderboard.** Following the process in Figure 1, we evaluate the performance of attacks vs. defenses in *graph injection* scenario. Table 3 shows an example of leaderboard for *grb-aminer* dataset. Each attack is repeated 10 times to report the error bar. Both attacks and defenses are ranked by the weighted accuracy under 'F' difficulty, where red and blue indicate the best results of attacks/defenses in each difficulty. Note that the metric is not fixed and will be updated when there are more effective methods. For instance, when there are more powerful attacks, the ranking will change so as the attached weights. It is reasonable that less effective attacks become less important on the final ranking of defenses, the same for defenses. As a result, GRB leaderboard can indicate the most robust defenses and the most effective attacks.

Table 3: *grb-aminer* leaderboard (Top 5 ATK. vs. Top 10 DEF.) in *graph injection* scenario.

| Attacks | | 1 GAT+AT | 2 R-GCN+AT | 3 SGCN+LN | 4 R-GCN | 5 GCN+LN | 6 GAT+LN | 7 GIN+LN | 8 TAGCN+LN | 9 TAGCN+AT | 10 GAT | Avg. Accuracy | Avg. 3-Max Accuracy | Weighted Accuracy |
|---|---|---|---|---|---|---|---|---|---|---|---|---|---|---|
| **1 TDGIA** | E | 59.54±0.05 | 56.83±0.06 | 56.73±0.06 | 56.12±0.07 | 53.51±0.21 | 43.93±0.41 | 51.10±0.12 | 54.63±0.20 | 49.59±0.50 | 42.40±0.52 | 52.44±0.17 | 57.70±1.31 | 58.08±0.04 |
| | M | 68.39±0.02 | 65.61±0.02 | 66.11±0.02 | 65.23±0.03 | 66.78±0.05 | 61.84±1.20 | 64.49±0.10 | 64.62±0.02 | 67.27±0.04 | 62.47±1.01 | 65.28±0.23 | 67.48±0.68 | 67.69±0.02 |
| | H | 75.83±0.02 | 72.35±0.02 | 72.10±0.00 | 71.94±0.02 | 73.39±0.02 | 75.22±0.04 | 72.92±0.02 | 68.94±0.03 | 73.98±0.01 | 75.03±0.03 | 73.17±0.01 | 75.36±0.34 | 75.33±0.01 |
| | F | 67.69±0.03 | 63.62±0.32 | 62.20±0.15 | 61.99±0.22 | 60.38±1.46 | 59.69±1.57 | 59.59±0.42 | 59.06±1.75 | 57.24±5.04 | 56.63±6.75 | 60.81±1.71 | 64.52±2.32 | 65.74±0.21 |
| **2 SPEIT** | E | 59.54±0.07 | 56.80±0.05 | 56.94±0.10 | 55.64±0.10 | 56.15±0.06 | 56.13±0.07 | 54.24±0.09 | 56.61±0.06 | 56.59±0.08 | 57.36±0.09 | 56.60±0.04 | 57.95±1.14 | 58.62±0.05 |
| | M | 68.37±0.03 | 65.46±0.03 | 66.20±0.02 | 65.25±0.05 | 66.75±0.03 | 67.49±0.06 | 65.05±0.06 | 64.47±0.04 | 66.95±0.05 | 66.81±0.04 | 66.28±0.04 | 67.60±0.59 | 67.86±0.03 |
| | H | 75.94±0.04 | 72.27±0.03 | 72.36±0.03 | 71.86±0.03 | 73.41±0.01 | 75.34±0.03 | 72.87±0.03 | 68.88±0.05 | 73.98±0.02 | 73.83±0.04 | 73.07±0.01 | 75.08±0.82 | 75.33±0.02 |
| | F | 68.04±0.03 | 64.05±0.04 | 64.84±0.04 | 64.06±0.04 | 65.51±0.02 | 64.02±0.04 | 63.11±0.02 | 62.59±0.04 | 63.77±0.06 | 63.58±0.05 | 64.36±0.02 | 66.13±1.38 | 66.89±0.02 |
| **3 RND** | E | 59.56±0.06 | 57.53±0.06 | 57.41±0.06 | 56.38±0.11 | 57.76±0.05 | 58.83±0.10 | 54.41±1.13 | 58.07±0.12 | 58.14±0.04 | 57.46±0.10 | 57.55±0.03 | 58.85±0.57 | 59.09±0.05 |
| | M | 68.22±0.04 | 65.86±0.03 | 66.29±0.03 | 65.34±0.06 | 67.03±0.03 | 68.62±0.05 | 65.54±0.06 | 64.98±0.08 | 67.34±0.04 | 67.71±0.06 | 66.69±0.02 | 68.18±0.38 | 68.24±0.03 |
| | H | 75.75±0.02 | 72.66±0.02 | 72.42±0.03 | 72.00±0.03 | 73.52±0.02 | 75.63±0.03 | 73.36±0.03 | 69.30±0.06 | 74.04±0.02 | 75.36±0.03 | 73.40±0.01 | 75.58±0.17 | 75.39±0.01 |
| | F | 67.72±0.04 | 64.98±0.02 | 65.31±0.04 | 64.45±0.04 | 66.17±0.02 | 67.54±0.04 | 64.36±0.06 | 64.33±0.03 | 66.42±0.03 | 66.23±0.04 | 65.75±0.02 | 67.23±0.58 | 67.34±0.03 |
| **4 PGD** | E | 59.70±0.06 | 57.71±0.05 | 57.73±0.09 | 57.19±0.07 | 57.60±0.08 | 57.05±0.17 | 54.69±0.09 | 58.18±0.07 | 58.27±0.09 | 58.46±0.11 | 57.66±0.05 | 58.81±0.64 | 59.14±0.05 |
| | M | 68.40±0.01 | 66.12±0.02 | 66.39±0.04 | 65.67±0.04 | 67.04±0.03 | 68.24±0.04 | 65.64±0.08 | 65.17±0.05 | 67.32±0.03 | 67.85±0.05 | 66.78±0.02 | 68.16±0.23 | 68.12±0.03 |
| | H | 75.83±0.03 | 72.91±0.02 | 72.47±0.04 | 72.18±0.05 | 73.52±0.02 | 75.55±0.05 | 73.58±0.04 | 69.64±0.05 | 73.89±0.02 | 74.34±0.04 | 73.39±0.01 | 75.24±0.65 | 75.36±0.02 |
| | F | 68.01±0.02 | 65.41±0.01 | 65.54±0.03 | 65.05±0.03 | 66.22±0.02 | 66.49±0.04 | 64.63±0.04 | 64.82±0.04 | 66.32±0.02 | 66.14±0.04 | 65.86±0.01 | 66.94±0.76 | 67.37±0.02 |
| **5 FGSM** | E | 59.71±0.05 | 57.69±0.08 | 57.62±0.06 | 57.16±0.08 | 57.60±0.06 | 56.97±0.09 | 54.67±0.08 | 58.20±0.10 | 58.23±0.06 | 58.46±0.07 | 57.63±0.05 | 58.81±0.65 | 59.15±0.04 |
| | M | 68.37±0.02 | 66.10±0.03 | 66.38±0.04 | 65.70±0.05 | 67.03±0.04 | 68.27±0.03 | 65.61±0.08 | 65.16±0.05 | 67.30±0.02 | 67.84±0.07 | 66.78±0.02 | 68.16±0.23 | 68.11±0.02 |
| | H | 75.82±0.02 | 72.92±0.04 | 72.48±0.03 | 72.18±0.05 | 73.52±0.02 | 75.55±0.05 | 73.60±0.04 | 69.64±0.04 | 73.90±0.01 | 74.34±0.04 | 73.39±0.01 | 75.23±0.65 | 75.35±0.02 |
| | F | 68.00±0.02 | 65.41±0.02 | 65.54±0.04 | 65.05±0.04 | 66.22±0.02 | 66.50±0.06 | 64.65±0.04 | 64.82±0.04 | 66.34±0.03 | 66.15±0.06 | 65.87±0.01 | 66.95±0.75 | 67.37±0.01 |
| **6 W/O Attack** | E | 59.67±0.00 | 58.08±0.00 | 60.22±0.00 | 58.53±0.00 | 58.14±0.00 | 60.78±0.00 | 56.83±0.00 | 58.20±0.00 | 59.62±0.00 | 59.88±0.00 | 59.12±0.00 | 60.29±0.37 | 60.42±0.00 |
| | M | 68.28±0.00 | 66.14±0.00 | 67.11±0.00 | 66.35±0.00 | 67.00±0.00 | 68.98±0.00 | 66.26±0.00 | 65.41±0.00 | 67.53±0.00 | 68.41±0.00 | 67.15±0.00 | 68.56±0.30 | 68.59±0.00 |
| | H | 75.85±0.00 | 73.05±0.00 | 72.69±0.00 | 72.66±0.00 | 73.46±0.00 | 75.64±0.00 | 73.69±0.00 | 69.84±0.00 | 74.10±0.00 | 75.76±0.00 | 73.67±0.00 | 75.75±0.09 | 75.52±0.00 |
| | F | 67.93±0.00 | 65.76±0.00 | 66.68±0.00 | 65.85±0.00 | 66.20±0.00 | 68.47±0.00 | 65.59±0.00 | 64.91±0.00 | 67.08±0.00 | 68.02±0.00 | 66.65±0.00 | 68.14±0.24 | 68.11±0.00 |
| **Avg. Accuracy** | E | 59.62±0.02 | 57.44±0.03 | 57.77±0.04 | 56.84±0.04 | 56.79±0.04 | 55.62±0.06 | 54.33±0.04 | 57.53±0.05 | 56.74±0.09 | 55.67±0.10 | - | - | - |
| | M | 68.34±0.01 | 65.88±0.01 | 66.41±0.01 | 65.59±0.02 | 66.94±0.02 | 67.24±0.19 | 65.43±0.03 | 64.97±0.02 | 67.28±0.01 | 66.85±0.18 | - | - | - |
| | H | 75.84±0.01 | 72.69±0.01 | 72.42±0.01 | 72.14±0.02 | 73.47±0.01 | 75.49±0.01 | 73.33±0.02 | 69.38±0.02 | 73.98±0.00 | 74.78±0.02 | - | - | - |
| | F | 67.90±0.01 | 64.87±0.05 | 65.02±0.04 | 64.41±0.04 | 65.12±0.25 | 65.45±0.26 | 63.65±0.07 | 63.42±0.09 | 64.53±0.04 | 64.46±1.13 | - | - | - |
| **Avg. 3-Min Accuracy** | E | 59.55±0.03 | 57.05±0.04 | 57.02±0.03 | 56.05±0.07 | 55.73±0.07 | 52.33±0.12 | 53.25±0.07 | 56.43±0.07 | 54.77±0.16 | 52.41±0.17 | - | - | - |
| | M | 68.28±0.01 | 65.64±0.02 | 66.20±0.01 | 65.28±0.03 | 66.84±0.02 | 65.85±0.40 | 65.02±0.04 | 64.69±0.03 | 67.17±0.02 | 65.66±0.34 | - | - | - |
| | H | 75.80±0.02 | 72.42±0.02 | 72.29±0.01 | 71.93±0.02 | 73.42±0.01 | 75.36±0.02 | 73.05±0.02 | 69.04±0.03 | 73.92±0.01 | 74.17±0.03 | - | - | - |
| | F | 67.78±0.02 | 64.22±0.11 | 64.12±0.06 | 63.50±0.08 | 63.39±0.53 | 62.35±0.14 | 61.99±0.58 | 62.44±1.69 | 62.11±2.26 | | - | - | - |
| **Weighted Accuracy** | E | 59.53±0.04 | 56.93±0.04 | 56.94±0.04 | 55.93±0.08 | 54.63±0.14 | 48.21±0.27 | 52.23±0.08 | 55.55±0.14 | 52.18±0.33 | 47.45±0.35 | - | - | - |
| | M | 68.25±0.02 | 65.57±0.02 | 66.17±0.02 | 65.28±0.02 | 66.79±0.02 | 63.85±0.80 | 64.77±0.07 | 64.80±0.04 | 67.06±0.03 | 64.07±0.68 | - | - | - |
| | H | 75.78±0.02 | 72.37±0.02 | 72.20±0.01 | 71.92±0.03 | 73.41±0.01 | 75.30±0.02 | 72.98±0.02 | 68.99±0.04 | 73.91±0.01 | 74.08±0.03 | - | - | - |
| | F | 67.73±0.03 | 63.96±0.21 | 63.19±0.10 | 62.80±0.15 | 62.18±0.98 | 61.58±1.05 | 61.00±0.28 | 60.54±1.18 | 59.82±3.38 | 59.37±4.53 | - | - | - |

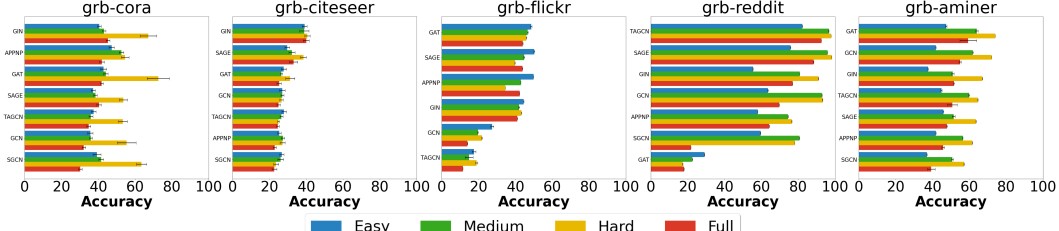

Figure 5: Ranking of vanilla GML models in *graph injection* scenario for all five datasets.

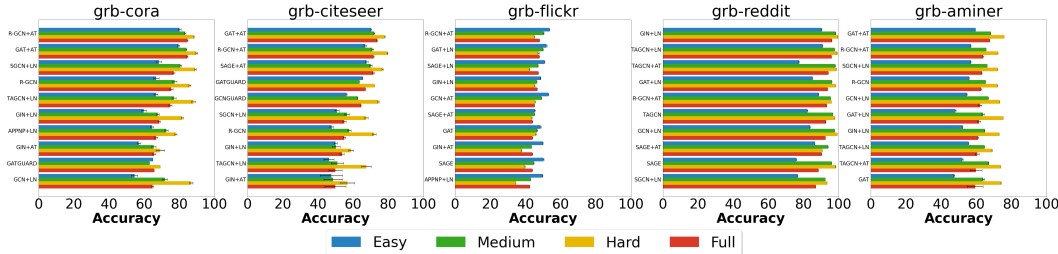

Figure 6: Ranking of top 10 defensed GML models in *graph injection* scenario for all five datasets.

**Robust Ranking of GML Models.** In Figure 5 and 6, we show the rankings of GML models for all five datasets in *graph injection* scenario. The ranking is determined by $s_w^{\text{DEF}}$ calculated by multiple attacks, which makes the results more general than previous works (only consider very few attacks and vanilla GML models). We find that the rankings are different across datasets, indicating that the

robustness is related to the properties of graph data. Similar situations can be found in other graph benchmarks. For example in OGB, there is no dominant GML model, the performance of certain model architecture may vary a lot across datasets. Thus, we suggest that *when giving conclusions about robustness in GML, one should not only consider the model itself but also take the graph data into account*. GRB provides scalable datasets of various domains, which can help to investigate the robustness of GML models in different situations. Among current vanilla GML models, we find that GAT and GIN generally perform better under attacks in several datasets, which might be due to the higher expressiveness of model architecture. Meanwhile, models like APPNP and SGCN that rely on high-order message propagation seem to be sensible to perturbations on the graph. Besides, GML models with defense mechanisms (*i.e.*, R-GCN, GNNGuard) are generally more robust. Moreover, we find simple methods like LN can be applied to all GML models to increase robustness. In the following, we further analyze some factors that affect the adversarial robustness of GML models.

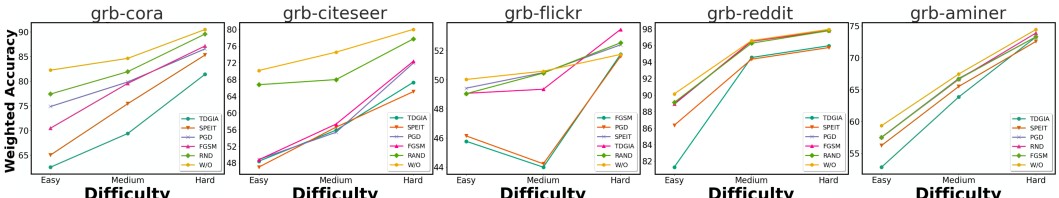

Figure 7: Effect of dataset difficulties on the performance of *graph injection* attacks.

**Effect of Difficulties.** The new splitting scheme investigates the effect of the average degree of target nodes on the attack performance. In Figure 7, attacks tend to better decrease the performance on nodes with lower degrees, which confirms the assumption that these low-degree nodes are more vulnerable. Moreover, according to Figure 5 and 6, the robustness on these nodes is indeed harder to achieve. This phenomenon encourages future work to deal with these vulnerable nodes to design more robust GML models.

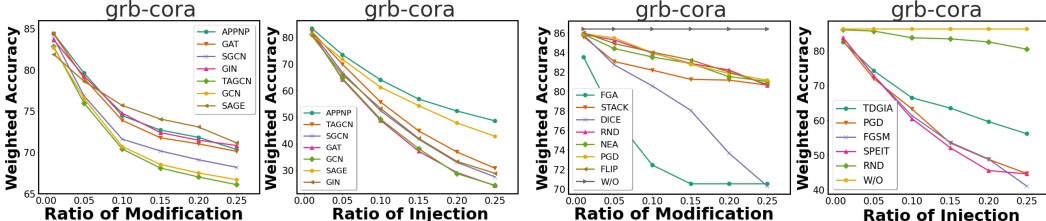

Figure 8: Effect of constraints on GML models. Figure 9: Effect of constraints on attacks. Left: Left: *graph modification*. Right: *graph injection*. *graph modification*. Right: *graph injection*.

**Effect of Constraints.** As shown in Figure 8 and 9, for both *graph modification* and *graph injection* scenarios, the variation of constraints on the ratio of modification/injection affects the effectiveness of attacks. Meanwhile, the ranking of methods nearly agrees with different constraints. Without loss of generality, it is reasonable to fix a specific constraint to build GRB leaderboards, where the relative robustness of GML models will still be indicative.

**Effect of General Defenses.** Figure 10 and 11 shows the results of the adapted LN and AT for all five datasets. LN is a node-wise normalization technique, which can alleviate the perturbations on node features as well as hidden features in each layer of GML models. AT applies adversarial attacks during training via modification or injection, which changes the decision boundary of models to tolerate perturbed nodes. The results indicate that these approaches can generally increase the robustness of various types of GML models, which can serve as simple but strong baselines for future works. The details of these algorithms can be found in Appendix A.4.3.

## 5 Conclusion

To improve and facilitate the evaluation of the adversarial robustness in GML, we revisit the limitations of previous works and present the Graph Robustness Benchmark (GRB), a *scalable*, *unified*, *modular*, and *reproducible* benchmark. It has scalable datasets, unified evaluation scenarios, as well as a

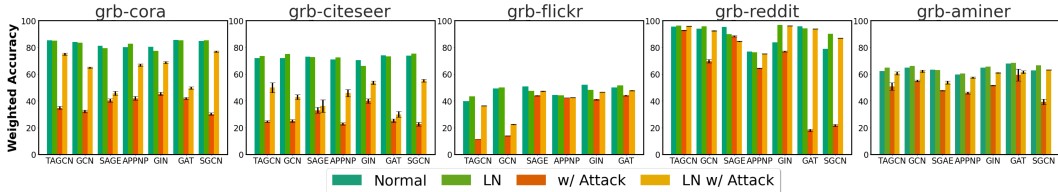

Figure 10: Effect of the adapted LN on the adversarial robustness of vanilla GML models for all five datasets. Adding LN can generally increase robustness of GML models.

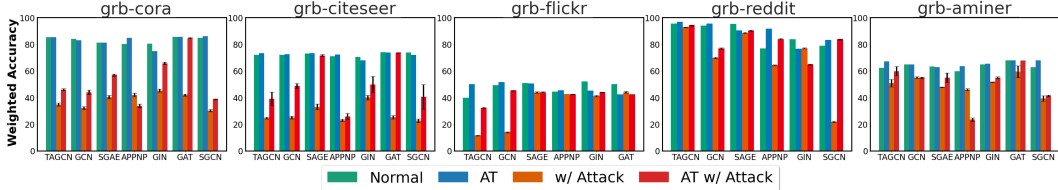

Figure 11: Effect of the adapted AT on the adversarial robustness of vanilla GML models for all five datasets. Adding AT can generally increase robustness of GML models.

modular coding framework that ensures the reproducibility and promotes the development of future methods. Extensive experiments with GRB provide insights on the understanding of the adversarial robustness in GML. We welcome the community to contribute more advanced GML models, attacks and defenses to further enrich GRB and to promote the research of this field.

## 6 Broader Impact

**Positive Impact.** GRB provides a general framework for GML attacks and defenses. On one hand, it will help researchers to develop more robust GML models against attacks. On the other hand, it will also help possible attackers to develop better attack methods to turn down defenses. More public information of potential attacks will make it harder to conduct secret attacks based on private methods. As a result, more generally robust defense mechanisms can be designed.

**Negative Impact.** By exposing the attack methods widely, the GML models may face more threats. Attackers can use the benchmark to design destructive attacks that may cause damage to GML-based systems. Additionally, GRB has some limitations. For example, it only considers homogeneous graphs rather than heterogeneous ones for now. It focuses on node classification, while other tasks like link prediction and graph classification are also vulnerable. We will regularly update GRB (*e.g.*, adding task-specific modules, designing related metrics.) to overcome these limitations.

## 7 Maintenance Plan

**Open Source.** We host the GRB homepage (https://cogdl.ai/grb/home) with detailed introduction, leaderboards, and documentations. The codes are available in (https://github.com/THUDM/grb). All materials are accessible to ensure reproducibility.

**Submissions of New Methods.** GRB will regularly include SOTA methods by updating the "method zoo". To welcome the contribution of the community, we allow submissions through google form. There are detailed examples and rules that guide researchers to add new attacks or defenses. Results will be updated on leaderboards to track the progress of the domain.

**Extension of Tasks.** Due to the modular design, GRB can be extended to other tasks. It requires adding task-specific functions in each module (dataset, model, trainer, attack, defense, etc.). Other common functions in GML can be reused for different tasks. There are online examples (https://github.com/THUDM/grb/tree/master/examples) showing how to use GRB for other tasks, *e.g.*, graph classification. In the future, GRB will support more GML tasks and define related threat models and metrics to unify the evaluation of adversarial robustness.

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
