# A Appendix

## A.1 Datasets

Table 4: Statistics of GRB datasets after new splitting scheme and feature normalization.

| Dataset | Splitting | Avg. Deg. | Avg.Deg. (E / M / H / F) | Feature range (original) | Feature range (normalized) |
|---------|-----------|-----------|--------------------------|--------------------------|----------------------------|
| *grb-cora* | **Train / Val** | 3.84 | 1.53/2.96/5.23/3.24 | [-2.30, 2.40] | [-0.94, 0.94] |
| *grb-citeseer* | 0.6 / 0.1 | 2.61 | 1.01/1.74/3.82/2.19 | [-4.55, 1.67] | [-0.96, 0.89] |
| *grb-flickr* | **Test: E / M / H / F** | 10.08 | 5.00/6.02/11.03/7.35 | [-0.90, 269.96] | [-0.47, 1.00] |
| *grb-reddit* | 0.1 / 0.1 / 0.1 / 0.3 | 99.65 | 29.23/68.36/150.99/82.86 | [-28.19, 120.96] | [-0.98, 0.99] |
| *grb-aminer* | | 8.73 | 1.99/5.12/13.25/6.79 | [-1.74, 1.62] | [-0.93, 0.93] |

GRB includes five datasets of different scales, the details of them are as following:

- *grb-cora*: Small-scale citation networks. Each node represents a research paper, and each edge represents a citation relationship between two papers. Instead of using the popular version of Cora used in Planetoid [44], we use a refined version [42], which removes duplicated nodes and generates indirect pre-trained word embeddings as node features to solve the problem of information leakage in the original version. As a result, the features become 302-dimension continuous features rather than 1433-dimension binary features in the original version. The task is to classify papers into 7 categories.

- *grb-citeseer*: Small-scale citation networks. Similar to *grb-cora*, we use a refined version [42] of CiteSeer, which eliminates identical papers and generates text embeddings by pre-trained BERT [45] model. The resulting features are 768-dimension continuous features rather than 3703-dimension binary features in the original version. The task is to classify papers into 6 categories.

- *grb-flickr*: Medium-scale social networks. We adopt the Flickr dataset from [43], which contains descriptions and common properties of online images. The dataset is processed with a new splitting scheme and feature normalization mentioned in 4. The task is to classify images into 7 categories.

- *grn-reddit*: Large-scale social networks. We adopt the Reddit dataset from [43], which contains the communities of online posts based on user comments. The task is to classify communities into 41 categories.

- *grb-aminer*: Large-scale citation networks. The papers are collected from the academic searching engine Aminer [46], and the dataset was used in KDD-CUP 2020 Graph Adversarial Attack & Defense competition. The task is to classify papers into 18 categories.

All five datasets are processed by the new splitting scheme and feature normalization mentioned in 4. The datasets are saved in the format of numpy [47] zipped format (with `.npz` extension), and each has four files: `adj.npz`, `features.npz`, `index.npz` and `labels.npz`. The data can be loaded by using the *Dataset* module in GRB. All data are maintained and can be found in https://cogdl.ai/grb/datasets, where we will continuously update to ensure the accessibility for a long term. We use MIT license for data and codes.

## A.2 Related Works

In other domains like image classification, there are already standards [48] or benchmarks [49] [50] for evaluating adversarial robustness. Besides, there exists a toolkit like DeepRobust [28] that implements adversarial attacks and defenses for both image classification and GML tasks. There are currently several benchmarks in GML. Open Graph Benchmark (OGB) [26] develops a diverse set of scalable and realistic datasets, which facilitates the evaluation of GML models. Dwivedi et al. [27] proposes a reproducible GNN benchmarking framework to facilitate researchers to add new models conveniently for arbitrary datasets. These benchmarks mainly focus on the performance but not the robustness of GNNs. so far, there is no benchmark on evaluating the *adversarial robustness* of GML models, i.e. the robustness in the presence of adversarial attacks. Nevertheless, it is an important but challenging task, which requires avoiding pitfalls in previous works and proposing a better solution.

## A.3 Rethinking Ill-defined Evaluation Scenarios in Previous Works

Many of the previous adversarial attacks [12, 30, 13] consider the *poisoning* attack and develop the notion of *unnoticeability*, similar to Eq. 1. The initial idea is to imitate the same notion in image classification task: the differences of adversarial examples, compared with clean examples, should be tiny and unnoticeable, so that humans can still easily recognize the objects in images. That's why $l_p$-norm is a widely-used constraint, as it corresponds to the visual sense of humans.

In the *poisoning* setting of graph modification attacks, the attackers assume that the graph is perturbed with corrupted nodes and edges, in a way that the perturbed graph is close to the original one. However, this assumption is controversial: If defenders have the original graph, they can simply train the model on that one; If defenders do not have the original graph (the general case for data poisoning where defenders can not tell whether the data are benign or not), then it does not make sense to keep *unnoticeability*. In this case, we only have $\mathcal{G}' = (\mathcal{A}', \mathcal{F}')$ but not $\mathcal{G} = (\mathcal{A}, \mathcal{F})$ in Eq. 1, making it almost impossible to compare them. Previous works propose to compare the graph properties, like degree distribution [12], feature statistics [33] or topological properties [20]. However, all these comparisons need to be done in presence of the original graph. This is different from the case of images, where *unnoticeability* can be easily judged by humans even without ground-truth images.

The attackers may perturb the graph structure or attributes within the scope of *unnoticeability* defined by themselves, while defenders have to depend on their own observations to discover. For example, Nettack [12] proposes to keep the degree distribution of modified graph similar to the original one. However, even if defenders notice that the degree distribution is different, it is still hard to identify specific malicious nodes or edges from the entire graph. On the contrary, defenses like GNNGuard [25] can use the dissimilarity between features to alleviate effects of perturbations. We argue that it is inadequate to simply adopt the notion from image classification, and to make two graphs "similar" in whatever way. Indeed, there is not an absolute definition, but it is recommended that: *"Unnoticeability" shall be considered from the defenders' view instead of the attackers'*.

As a starting point, we consider very basic constraints in GRB (*e.g.*, a limited number of modified edges or nodes). Pre-defined complex constraints might ignore the real capability of attackers and defenders, and might be obsoleted as the research goes. Thus, we do not add too many constraints and we insist that the notion like "unnoticeability" will be refined during the arms race between attackers and defenders. For example, if an advanced defense proposes a measure to identify malicious nodes with high probability, then the attackers can decide by themselves to refine the constraints based on this measure. There will be a trade-off, considering more constraints for one specific defense might result in less effectiveness for other methods. That's also why GRB considers a general metric across multiple attacks/defenses rather than a single pair of attack/defense. We insist that finding methods that are generally more effective bring much more value in practical applications.

## A.4 Methodology

### A.4.1 GML Models

GCN (Graph Convolutional Networks) [4] introduces a layer-wise propagation rule for graph-structured data which is motivated from a first-order approximation of spectral graph convolutions. GAT (Graph Attention Networks) [6] leverages masked self-attention layers where nodes can attend over their neighborhoods' features with different weights. GIN (Graph Isomorphic Networks) [7] is a theoretically guaranteed framework for analyzing the expressive power of GNNs to capture different graph structures. APPNP (Approximated Personalized Propagation of Neural Predictions ) [8] utilizes an improved propagation scheme based on personalized PageRank to construct a simple model with fast approximation. TAGCN (Topological Adaptive Graph Convolutional Networks) [20] provides a systematic way to design a set of fixed-size learnable filters to perform convolutions on graphs. GraphSAGE [5] is a general inductive framework that leverages node features to generate node embeddings for previously unseen data. SGCN (Simplified Graph Convolutional Networks) [9] removes nonlinearities and collapses weight matrices between consecutive layers, resulting in a linear model.

### A.4.2 Adversarial Attacks

**Modification Attacks.** RND (Random) [12] is a random attack strategy that only modifies the structure of the graph. DICE (Delete Internally Connect Externally) [19] deletes edges with the label, and adds edges with different labels. FGA (Fast Gradient Attack) [11] calculates the gradient of dense adjacency matrix related to the classification loss and identifies the most vulnerable edges to perturb. FLIP (Flipping attack) [29] is a deterministic approach that first ranks all nodes in ascending order according to their degrees, then flips their edges from the lower degree nodes to higher degree nodes. NEA (Network Embedding Attack) [29] is a black-box attack originally designed for attacking Deepwalk. STACK (Strict Black-box Attack) [31] does not require any knowledge of the target model and does not need training a surrogate model. It uses a generic graph filter unifying different GML models as an estimation for applying optimization attacks. PGD (Projected Gradient Descent) [34] is originally an attack only modifying the features of inputs. Here, we adapt it to first randomly perturb edges of the graph, then optimize the features of target nodes by projected gradient descent.

**Injection Attacks.** In *graph injection* scenario, RND (Random) refers to randomly injecting new nodes with features randomly generated from the Gaussian distribution. FGSM (Fast Gradient Sign Method) [40] linearizes the cost function around the current value of parameters, obtaining an optimal max-norm constrained perturbation, which is called the "fast gradient sign method" of generating adversarial examples. We use an iterative version of FGSM to conduct a graph injection attack. PGD (Projected Gradient Descent) [34] is a universal "first-order adversary", *i.e.*, the strongest attack utilizing the local first-order information about the network. The feature initialization is different from FGSM. SPEIT [17] is the first place solution in KDD-CUP 2020 Graph Adversarial Attack & Defense competition. It consists of adversarial adjacent matrix generation and enhanced feature gradient attacks, which are designed as a universal black-box graph injection attack. TDGIA (Topological Defective Graph Injection Attack) [18] is an effective graph injection attack that tackles the topological defectiveness of graphs. By sequentially injecting malicious nodes around nodes that are topologically vulnerable, TDGIA can significantly influence the accuracy of GML models.

Since the proposed scenario in GRB is a *black-box* one, all the above attacks are first applied to a surrogate model (trained by the attackers themselves), and then transfer to the target model. As demonstrated in [18], the choice of surrogate model will influence the transferability of attacks. When using raw GCN as the surrogate model, attacks can generally achieve better performance. Thus in GRB experiments, we use GCN as the surrogate model for all attacks. Nevertheless, we encourage future researchers to further investigate the effect of transferability by testing other methods.

### A.4.3 Defenses

GNN-SVD [24] utilizes a low-rank approximation of the graph, that uses only the top singular components for its reconstruction. GNN-Guard [25] introduces the neighbor importance estimation and the layer-wise graph memory for defenses. RobustGCN (R-GCN) [51] is a GCN variant that is specially designed against adversarial attacks on graphs. It adapts the random perturbation of features from VAE [52] and encodes both the mean and variance of the node representation thus makes the GNNs more robust. However, we found that methods like GNN-SVD and GNNGuard are not scalable to large-scale graphs due to the calculation of large dense matrices. To have stronger baseline defenses, we propose two methods that are scalable and can generally improve the performance of GML models.

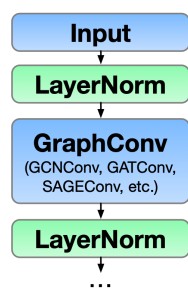

Figure 12: The proposed layer normalization in GRB. It is applied on the input and after every graph convolutional layer except the last one.

**The adapted layer normalization (LN).** LN [41] computes the mean and variance used for normalization from all of the summed inputs to the neurons in a layer on a single training case. It is originally used to stabilize the hidden state dynamics in recurrent networks. We found that it can also help to improve the adversarial robustness of GML models. Unlike the original version that is only used after hidden layers, we use LN first on the input features, and then after every graph convolutional layer except the last one. The process of the proposed LN is illustrated in Figure 12. The experiment

results in Section 4 show that the proposed LN can generally improve the adversarial robustness of different types of GML models.

**The adapted adversarial training (AT) in *graph injection* scenario.** The AT [34] is originally designed for defending adversarial attacks in image classification. The idea is to generate adversarial examples during training to change the classification distribution of models, which makes it difficult to perturb the results. Previous works [35] show that AT is also helpful for GML models, but it only considers the problem of graph modification attack, where the original graph can be modified. In our case, the defense is against graph injection attack, thus we propose a variant of AT that conduct graph injection attack during training. The procedure of the proposed AT is as following: (1) Initialization: the training graph is first used to train GML models for a few iterations as a warm-up. (2) FGSM attack: we conduct FGSM attack for a few steps on the current model to inject malicious nodes to attack training nodes. (3) Update gradients: we then train on the injected graph and minimize the classification loss of training nodes (excluding the injected nodes). (4) Repetition: we repeat this AT process until the training loss converges. Finally, we are able to construct more robust GML models. Interestingly, we found that AT with FGSM can also defend against other attacks, which shows great generality. Besides, the proposed AT can be easily adapt to any kind of GML models and scalable to large graphs.

Table 5: Hyper-parameters for adversarial training for five datasets.

| Dataset | Attack | Step size | # Steps/Iter | # Injection | # Edges | Feature range |
|---------|--------|-----------|--------------|-------------|---------|---------------|
| *grb-cora* | FGSM | 0.01 | 10 | 20 | 20 | [-0.94, 0.94] |
| *grb-citeseer* | FGSM | 0.01 | 10 | 30 | 20 | [-0.96, 0.89] |
| *grb-flickr* | FGSM | 0.01 | 10 | 200 | 100 | [-0.47, 0.99] |
| *grb-reddit* | FGSM | 0.01 | 10 | 500 | 200 | [-0.98, 0.99] |
| *grb-aminer* | FGSM | 0.01 | 10 | 500 | 100 | [-0.93, 0.93] |

## A.5 Reproducibility

Reproducibility is one of the main features of GRB. For reproducing results on leaderboards, all necessary components are available, including model weights, attack parameters, generated adversarial results, etc. Besides, GRB provides scripts that allow users to reproduce results by a single command line. All codes are available in `https://github.com/THUDM/grb`, where the implementation details and examples can be found. GRB also provides full documentation for each module and function. All experiments can be reproduced in a single NVIDIA V100 GPU (with 32 GB memory).

### A.5.1 Hyper-Parameter Settings

**Hyper-Parameters of GML Models and Defenses.** The hyper-parameters of vanilla GML models and defenses are shown in Table 6, 7, 8, 9, 10, where GCNGuard stands for GCN+GNNGuard, GATGuard for GAT+GNNGuard, GCN-SVD for GCN+GNN-SVD, and LN for the proposed layer normalization. For the proposed adversarial training (AT), the hyper-parameters are shown in Table 5. Under the proposed AT, GML models are trained while being continuously attacked by FGSM attack for a few steps per training iterations. Note that in each iteration, the attack is independent of previous iterations, only based on the weights of the model in the current iteration. The other hyper-parameters are exactly the same as training GML models.

**Hyper-Parameters for Adversarial Attacks.** The hyper-parameters of attacks are shown in Table 12. For graph modification, following the most common setting in previous works, attackers are allowed to perturb a limited number of edges in the graph ($\Delta_{\mathcal{A}}$: the number of modified edges less than a ratio $\gamma_e$ of all edges). For graph injection, we follow the heuristic setting of KDDCUP 2020, attackers are allowed to inject new nodes with limited edges ($\Delta_{\mathcal{A}}$: less than $N_n$ injected nodes each with less than $N_e$ edges; $\Delta_{\mathcal{F}}$: constrained range of features $[\mathcal{F}_{min}, \mathcal{F}_{max}]$.). Nevertheless, more definitions of *unnoticeabilty* can be developed by attackers and defenders when using GRB. Since the proposed scenario in GRB is a *black-box* one, all the above attacks are first applied to a surrogate model (trained by the attackers themselves), and then transfer to the target model. As demonstrated in [18], the choice of surrogate model will influence the transferability of attacks. When using raw GCN as the surrogate model, attacks can generally achieve better performance. Thus in GRB experiments, we use GCN as the surrogate model for all attacks.

Table 6: Hyper-parameters of GML models for *grb-cora* dataset.

| Model | #Params | Hidden sizes | LR | Dropout | Optimizer | Others |
|---|---|---|---|---|---|---|
| GCN | 28,167 | 64, 64, 64 | 0.01 | 0.5 | Adam | |
| GCN+LN | 29,027 | 64, 64, 64 | 0.01 | 0.5 | Adam | |
| SAGE | 160,320 | 64, 64, 64 | 0.01 | 0.5 | Adam | full-batch |
| SAGE+LN | 161,180 | 64, 64, 64 | 0.01 | 0.5 | Adam | full-batch |
| SGCN | 28,771 | 64, 64, 64 | 0.01 | 0.5 | Adam | k=4 |
| SGCN+LN | 29,027 | 64, 64, 64 | 0.01 | 0.5 | Adam | k=4 |
| R-GCN | 56,334 | 64, 64, 64 | 0.01 | 0.5 | Adam | |
| TAGCN | 84,103 | 64, 64, 64 | 0.01 | 0.5 | Adam | k=2 |
| TAGCN+LN | 84,963 | 64, 64, 64 | 0.01 | 0.5 | Adam | k=2 |
| GAT | 217,940 | 64, 64, 64 | 0.01 | 0.5 | Adam | num_heads=4 |
| GAT+LN | 219,568 | 64, 64, 64 | 0.01 | 0.5 | Adam | num_heads=4 |
| APPNP | 19,847 | 64 | 0.01 | 0.5 | Adam | alpha=0.01, k=10 |
| APPNP+LN | 20,579 | 64 | 0.01 | 0.5 | Adam | alpha=0.01, k=10 |
| GIN | 45,194 | 64, 64, 64 | 0.01 | 0.5 | Adam | |
| GIN+LN | 46,054 | 64, 64, 64 | 0.01 | 0.5 | Adam | |
| GCNGuard | 24,010 | 64, 64 | 0.001 | 0.1 | Adam | |
| GATGuard | 151,639 | 64, 64 | 0.001 | 0.1 | Adam | num_heads=4 |
| GCN-SVD | 24,007 | 64, 64 | 0.01 | 0.5 | Adam | |

Table 7: Hyper-parameters of GML models for *grb-citeseer* dataset.

| Model | #Params | Hidden sizes | LR | Dropout | Optimizer | Others |
|---|---|---|---|---|---|---|
| GCN | 57,926 | 64, 64, 64 | 0.01 | 0.5 | Adam | |
| GCN+LN | 59,718 | 64, 64, 64 | 0.01 | 0.5 | Adam | |
| SAGE | 718,924 | 64, 64, 64 | 0.01 | 0.5 | Adam | full batch |
| SAGE+LN | 720,716 | 64, 64, 64 | 0.01 | 0.5 | Adam | full batch |
| SGCN | 59,462 | 64, 64, 64 | 0.01 | 0.5 | Adam | k=4 |
| SGCN+LN | 59,718 | 64, 64, 64 | 0.01 | 0.5 | Adam | k=4 |
| R-GCN | 115,852 | 64, 64, 64 | 0.01 | 0.5 | Adam | |
| TAGCN | 173,382 | 64, 64, 64 | 0.01 | 0.5 | Adam | k=2 |
| TAGCN+LN | 175,174 | 64, 64, 64 | 0.01 | 0.5 | Adam | k=2 |
| GAT | 336,200 | 64, 64, 64 | 0.01 | 0.5 | Adam | num_heads=4 |
| GAT+LN | 338,760 | 64, 64, 64 | 0.01 | 0.5 | Adam | num_heads=4 |
| APPNP | 49,606 | 64 | 0.01 | 0.5 | Adam | alpha=0.01, k=10 |
| APPNP+LN | 51,270 | 64 | 0.01 | 0.5 | Adam | alpha=0.01, k=10 |
| GIN | 74,953 | 64, 64, 64 | 0.01 | 0.5 | Adam | |
| GIN+LN | 76,745 | 64, 64, 64 | 0.01 | 0.5 | Adam | |
| GCNGuard | 53,769 | 64, 64 | 0.001 | 0.1 | Adam | |
| GATGuard | 269,899 | 64, 64 | 0.001 | 0.1 | Adam | num_heads=4 |
| GCN-SVD | 53,766 | 64, 64 | 0.01 | 0.5 | Adam | |

Table 8: Hyper-parameters of GML models for *grb-flickr* dataset.

| Model | #Params | Hidden sizes | LR | Dropout | Optimizer | Others |
|---|---|---|---|---|---|---|
| GCN | 169,863 | 256, 128, 64 | 0.01 | 0.5 | Adam | |
| GCN+LN | 171,631 | 256, 128, 64 | 0.01 | 0.5 | Adam | |
| SAGE | 496,146 | 128, 128, 128 | 0.01 | 0.5 | Adam | full batch |
| SAGE+LN | 497,658 | 128, 128, 128 | 0.01 | 0.5 | Adam | full batch |
| R-GCN | 196,110 | 128, 128, 128 | 0.01 | 0.5 | Adam | |
| TAGCN | 293,383 | 128, 128, 128 | 0.01 | 0.5 | Adam | k=2 |
| TAGCN+LN | 294,895 | 128, 128, 128 | 0.01 | 0.5 | Adam | k=2 |
| GAT | 799,316 | 128, 128, 128 | 0.01 | 0.5 | Adam | num_heads=4 |
| GAT+LN | 802,364 | 128, 128, 128 | 0.01 | 0.5 | Adam | num_heads=4 |
| APPNP | 65,031 | 128 | 0.01 | 0.5 | Adam | alpha=0.01, k=10 |
| APPNP+LN | 66,287 | 128 | 0.01 | 0.5 | Adam | alpha=0.01, k=10 |
| GIN | 164,874 | 128, 128, 128 | 0.01 | 0.5 | Adam | |
| GIN+LN | 166,386 | 128, 128, 128 | 0.01 | 0.5 | Adam | |
| GCNGuard | 81,546 | 128, 128 | 0.001 | 0.1 | Adam | |

Table 9: Hyper-parameters of GML models for *grb-reddit* dataset.

| Model | #Params | Hidden sizes | LR | Dropout | Optimizer | Others |
|---|---|---|---|---|---|---|
| GCN | 115,497 | 128, 128, 128 | 0.01 | 0.5 | Adam | |
| GCN+LN | 117,213 | 128, 128, 128 | 0.01 | 0.5 | Adam | |
| SAGE | 643,536 | 128, 128, 128 | 0.01 | 0.5 | Adam | full batch |
| SAGE+LN | 645,252 | 128, 128, 128 | 0.01 | 0.5 | Adam | full batch |
| SGCN | 116,701 | 128, 128, 128 | 0.01 | 0.5 | Adam | k=4 |
| SGCN+LN | 117,213 | 128, 128, 128 | 0.01 | 0.5 | Adam | k=4 |
| R-GCN | 230,994 | 128, 128, 128 | 0.01 | 0.5 | Adam | |
| TAGCN | 345,641 | 128, 128, 128 | 0.01 | 0.5 | Adam | k=2 |
| TAGCN+LN | 347,357 | 128, 128, 128 | 0.01 | 0.5 | Adam | k=2 |
| GAT | 104,950 | 64, 64 | 0.01 | 0.5 | Adam | num_heads=2 |
| GAT+LN | 106,410 | 64, 64 | 0.01 | 0.5 | Adam | num_heads=2 |
| APPNP | 82,473 | 128 | 0.01 | 0.5 | Adam | alpha=0.01, k=10 |
| APPNP+LN | 83,933 | 128 | 0.01 | 0.5 | Adam | alpha=0.01, k=10 |
| GIN | 182,316 | 128, 128, 128 | 0.01 | 0.5 | Adam | |
| GIN+LN | 184,032 | 128, 128, 128 | 0.01 | 0.5 | Adam | |

Table 10: Hyper-parameters of GML models for *grb-aminer* dataset.

| Model | #Params | Hidden sizes | LR | Dropout | Optimizer | Others |
|---|---|---|---|---|---|---|
| GCN | 48,274 | 128, 128, 128 | 0.01 | 0.5 | Adam | |
| GCN+LN | 48,986 | 128, 128, 128 | 0.01 | 0.5 | Adam | |
| SAGE | 156,184 | 128, 128, 128 | 0.01 | 0.5 | Adam | full batch |
| SAGE+LN | 156,896 | 128, 128, 128 | 0.01 | 0.5 | Adam | full batch |
| SGCN | 48,474 | 128, 128, 128 | 0.01 | 0.5 | Adam | k=4 |
| SGCN+LN | 48,986 | 128, 128, 128 | 0.01 | 0.5 | Adam | k=4 |
| R-GCN | 96,548 | 128, 128, 128 | 0.01 | 0.5 | Adam | |
| TAGCN | 144,018 | 128, 128, 128 | 0.01 | 0.5 | Adam | k=2 |
| TAGCN+LN | 144,730 | 128, 128, 128 | 0.01 | 0.5 | Adam | k=2 |
| GAT | 177,624 | 64, 64, 64 | 0.01 | 0.5 | Adam | num_heads=2 |
| GAT+LN | 178,848 | 64, 64, 64 | 0.01 | 0.5 | Adam | num_heads=2 |
| APPNP | 15,250 | 128 | 0.01 | 0.5 | Adam | alpha=0.01, k=10 |
| APPNP+LN | 15,706 | 128 | 0.01 | 0.5 | Adam | alpha=0.01, k=10 |
| GIN | 115,093 | 128, 128, 128 | 0.01 | 0.5 | Adam | |
| GIN+LN | 115,805 | 128, 128, 128 | 0.01 | 0.5 | Adam | |

Table 11: Runtime (/s) of *graph injection* attacks on large-scale graphs.

| | Difficulty | RAND | FGSM | PGD | SPEIT | TDGIA |
|---|---|---|---|---|---|---|
| *grb-reddit* | F | 10.32 | 1110.92 | 1112.95 | 1119.10 | 6892.12 |
| | H | 10.14 | 243.32 | 244.10 | 263.65 | 3179.26 |
| | M | 9.97 | 1067.18 | 801.40 | 950.90 | 3126.48 |
| | E | 11.64 | 304.03 | 305.44 | 319.04 | 4053.48 |
| *grb-aminer* | F | 12.93 | 954.53 | 953.14 | 961.66 | 4079.35 |
| | H | 12.87 | 212.41 | 213.44 | 233.08 | 2673.79 |
| | M | 11.62 | 932.04 | 926.10 | 372.64 | 2612.44 |
| | E | 12.61 | 218.53 | 219.93 | 239.57 | 2640.93 |

## A.6 Detailed Experiment Results

We conduct extensive experiments on all datasets and build a leaderboard for each dataset. Here we show the results of *graph injection* scenario with Top-5 attacks vs. Top-10 defenses, full leaderboards can be found in https://cogdl.ai/grb/leaderboard. Both attacks and defenses are ranked by the weighted accuracy, where red and blue indicated the best results in each difficulty. It can be seen that different methods vary performance in different datasets. And it is also hard for attacks to be generally effective, especially in the presence of the proposed strong defense baselines. The runtime of attacks on large-scale graphs (*grb-aminer*, *grb-reddit*) can be found in Table 11.

Table 12: Hyper-parameters for attacks for five datasets in *graph injection* scenario.

| Dataset | Attack | Step size | # Iter | # Injection (E/M/H/F) | # Edges | Feature Range | Others |
|---|---|---|---|---|---|---|---|
| *grb-cora* | RND | - | 1 | | | | Random features |
| | PGD | 0.01 | 1000 | | | | - |
| | FGSM | 0.01 | 1000 | 20/20/20/60 | 20 | [-0.94, 0.94] | - |
| | SPEIT | 0.01 | 1000 | | | | - |
| | TDGIA | 0.01 | 1000 | | | | Sequential |
| *grb-citeseer* | RND | - | 1 | | | | Random features |
| | PGD | 0.01 | 1000 | | | | - |
| | FGSM | 0.01 | 1000 | 30/30/30/90 | 20 | [-0.96, 0.89] | - |
| | SPEIT | 0.01 | 1000 | | | | - |
| | TDGIA | 0.01 | 1000 | | | | Sequential |
| *grb-flickr* | RND | - | 1 | | | | Random features |
| | PGD | 0.01 | 2000 | | | | - |
| | FGSM | 0.01 | 2000 | 200/200/200/600 | 100 | [-0.47, 0.99] | - |
| | SPEIT | 0.01 | 2000 | | | | - |
| | TDGIA | 0.01 | 2000 | | | | Sequential |
| *grb-reddit* | RND | - | 1 | | | | Random features |
| | PGD | 0.01 | 2000 | | | | - |
| | FGSM | 0.01 | 2000 | 500/500/500/1500 | 200 | [-0.98, 0.99] | - |
| | SPEIT | 0.01 | 2000 | | | | - |
| | TDGIA | 0.01 | 2000 | | | | Sequential |
| *grb-aminer* | RND | - | 1 | | | | Random features |
| | PGD | 0.01 | 5000 | | | | - |
| | FGSM | 0.01 | 5000 | 500/500/500/1500 | 100 | [-0.93, 0.93] | - |
| | SPEIT | 0.01 | 5000 | | | | - |
| | TDGIA | 0.01 | 5000 | | | | Sequential |

Table 13: GRB leaderboard (Top 5 Attacks vs. Top 10 Defenses) for *grb-cora* dataset.

| Attack | | 1 R-GCN+AT | 2 GAT+AT | 3 SGCN+LN | 4 R-GCN | 5 TAGCN+LN | 6 GIN+LN | 7 APPNP+LN | 8 GIN+AT | 9 GATGuard | 10 GCN+LN | Avg. Accuracy | Avg. 3-Max Accuracy | Weighted Accuracy |
|---|---|---|---|---|---|---|---|---|---|---|---|---|---|---|
| 1 SPEIT | E | $79.97_{\pm1.56}$ | $80.97_{\pm1.23}$ | $66.16_{\pm1.96}$ | $74.62_{\pm2.80}$ | $63.28_{\pm1.76}$ | $55.60_{\pm2.13}$ | $71.57_{\pm0.86}$ | $54.48_{\pm1.76}$ | $64.93_{\pm0.00}$ | $48.06_{\pm2.38}$ | $52.01_{\pm0.09}$ | $53.48_{\pm0.25}$ | $53.44_{\pm0.17}$ |
| | M | $84.11_{\pm0.38}$ | $84.55_{\pm0.44}$ | $80.60_{\pm1.18}$ | $81.16_{\pm1.39}$ | $74.59_{\pm1.86}$ | $64.10_{\pm1.05}$ | $75.67_{\pm1.12}$ | $63.36_{\pm1.39}$ | $63.06_{\pm0.00}$ | $69.18_{\pm2.24}$ | $47.31_{\pm0.09}$ | $50.35_{\pm1.25}$ | $50.89_{\pm0.24}$ |
| | H | $88.51_{\pm0.80}$ | $89.78_{\pm1.00}$ | $89.74_{\pm0.71}$ | $88.47_{\pm0.54}$ | $89.85_{\pm0.55}$ | $80.37_{\pm1.10}$ | $80.64_{\pm1.14}$ | $63.51_{\pm3.22}$ | $69.03_{\pm0.00}$ | $88.55_{\pm1.32}$ | $42.72_{\pm0.07}$ | $47.34_{\pm2.27}$ | $48.44_{\pm0.12}$ |
| | F | $85.21_{\pm0.41}$ | $85.35_{\pm0.19}$ | $75.65_{\pm0.87}$ | $79.85_{\pm0.48}$ | $71.73_{\pm1.14}$ | $64.75_{\pm0.62}$ | $73.11_{\pm0.76}$ | $63.05_{\pm1.37}$ | $65.67_{\pm0.00}$ | $59.45_{\pm0.64}$ | $45.86_{\pm0.05}$ | $48.56_{\pm0.99}$ | $48.95_{\pm0.12}$ |
| 2 TDGIA | E | $81.68_{\pm1.81}$ | $80.52_{\pm0.72}$ | $72.05_{\pm3.19}$ | $68.13_{\pm3.50}$ | $73.36_{\pm3.86}$ | $68.77_{\pm2.56}$ | $64.18_{\pm1.71}$ | $63.02_{\pm3.37}$ | $64.93_{\pm0.00}$ | $65.93_{\pm4.07}$ | $50.12_{\pm0.14}$ | $53.22_{\pm1.01}$ | $53.30_{\pm0.12}$ |
| | M | $83.96_{\pm0.53}$ | $84.25_{\pm0.40}$ | $81.49_{\pm0.71}$ | $77.57_{\pm1.86}$ | $82.17_{\pm3.29}$ | $73.58_{\pm2.81}$ | $74.44_{\pm1.52}$ | $70.19_{\pm1.84}$ | $63.06_{\pm0.00}$ | $76.34_{\pm2.16}$ | $47.21_{\pm0.06}$ | $51.58_{\pm0.22}$ | $51.06_{\pm0.10}$ |
| | H | $88.21_{\pm0.18}$ | $90.30_{\pm0.00}$ | $88.92_{\pm0.80}$ | $86.83_{\pm0.85}$ | $87.39_{\pm1.81}$ | $84.52_{\pm1.28}$ | $78.58_{\pm1.30}$ | $80.63_{\pm3.06}$ | $69.03_{\pm0.00}$ | $86.64_{\pm1.63}$ | $45.17_{\pm0.07}$ | $48.53_{\pm1.14}$ | $48.68_{\pm0.06}$ |
| | F | $84.43_{\pm0.27}$ | $84.55_{\pm0.50}$ | $77.39_{\pm1.05}$ | $74.58_{\pm1.76}$ | $79.67_{\pm1.53}$ | $76.14_{\pm1.80}$ | $68.16_{\pm2.10}$ | $70.51_{\pm1.58}$ | $65.67_{\pm0.00}$ | $72.58_{\pm2.71}$ | $46.24_{\pm0.04}$ | $49.75_{\pm0.64}$ | $49.73_{\pm0.09}$ |
| 3 PGD | E | $83.02_{\pm1.26}$ | $80.60_{\pm1.04}$ | $73.88_{\pm2.41}$ | $67.80_{\pm2.24}$ | $74.78_{\pm2.36}$ | $70.07_{\pm1.79}$ | $66.19_{\pm1.50}$ | $62.65_{\pm1.33}$ | $64.93_{\pm0.00}$ | $68.58_{\pm3.00}$ | $50.11_{\pm0.11}$ | $53.19_{\pm0.98}$ | $53.29_{\pm0.14}$ |
| | M | $83.84_{\pm1.15}$ | $84.81_{\pm0.67}$ | $82.20_{\pm1.18}$ | $78.51_{\pm1.57}$ | $83.36_{\pm1.39}$ | $77.35_{\pm1.02}$ | $73.21_{\pm1.83}$ | $69.25_{\pm2.18}$ | $63.06_{\pm0.00}$ | $77.80_{\pm1.26}$ | $47.24_{\pm0.08}$ | $51.64_{\pm0.20}$ | $51.11_{\pm0.09}$ |
| | H | $88.88_{\pm0.50}$ | $90.48_{\pm0.67}$ | $89.96_{\pm0.75}$ | $85.86_{\pm0.83}$ | $89.48_{\pm1.07}$ | $85.97_{\pm0.82}$ | $78.47_{\pm1.29}$ | $80.82_{\pm0.89}$ | $69.03_{\pm0.00}$ | $88.32_{\pm0.56}$ | $45.18_{\pm0.05}$ | $48.50_{\pm1.14}$ | $48.68_{\pm0.06}$ |
| | F | $85.60_{\pm0.38}$ | $85.43_{\pm0.34}$ | $81.90_{\pm0.90}$ | $76.77_{\pm0.74}$ | $83.22_{\pm0.60}$ | $78.58_{\pm0.51}$ | $66.49_{\pm0.68}$ | $71.30_{\pm0.94}$ | $65.67_{\pm0.00}$ | $78.21_{\pm0.68}$ | $46.26_{\pm0.04}$ | $49.81_{\pm0.89}$ | $49.83_{\pm0.08}$ |
| 4 FGSM | E | $82.35_{\pm0.95}$ | $80.19_{\pm0.97}$ | $73.95_{\pm2.46}$ | $67.01_{\pm1.40}$ | $74.03_{\pm1.72}$ | $70.48_{\pm1.60}$ | $64.67_{\pm1.71}$ | $62.46_{\pm2.30}$ | $64.93_{\pm0.00}$ | $68.39_{\pm1.27}$ | $52.72_{\pm0.03}$ | $54.71_{\pm0.33}$ | $54.71_{\pm0.07}$ |
| | M | $84.25_{\pm1.24}$ | $85.11_{\pm0.64}$ | $82.46_{\pm0.93}$ | $77.87_{\pm1.43}$ | $84.29_{\pm1.02}$ | $78.77_{\pm1.09}$ | $73.10_{\pm1.13}$ | $69.74_{\pm1.60}$ | $63.06_{\pm0.00}$ | $78.47_{\pm1.30}$ | $48.71_{\pm0.16}$ | $51.81_{\pm0.67}$ | $51.93_{\pm0.09}$ |
| | H | $89.22_{\pm0.59}$ | $90.59_{\pm0.40}$ | $90.30_{\pm0.50}$ | $86.23_{\pm1.16}$ | $89.55_{\pm0.67}$ | $86.08_{\pm0.78}$ | $78.80_{\pm1.45}$ | $80.93_{\pm0.91}$ | $69.03_{\pm0.00}$ | $87.13_{\pm0.96}$ | $43.58_{\pm0.08}$ | $48.20_{\pm1.74}$ | $48.84_{\pm0.08}$ |
| | F | $85.01_{\pm0.41}$ | $85.33_{\pm0.72}$ | $81.53_{\pm1.27}$ | $76.62_{\pm0.90}$ | $83.00_{\pm0.61}$ | $78.26_{\pm0.88}$ | $67.09_{\pm1.18}$ | $71.64_{\pm0.78}$ | $65.67_{\pm0.00}$ | $77.60_{\pm1.14}$ | $48.26_{\pm0.03}$ | $51.45_{\pm0.77}$ | $51.61_{\pm0.06}$ |
| 5 RND | E | $82.28_{\pm0.93}$ | $80.26_{\pm1.24}$ | $76.57_{\pm2.44}$ | $73.54_{\pm1.44}$ | $78.36_{\pm1.74}$ | $68.81_{\pm0.82}$ | $67.95_{\pm1.68}$ | $67.69_{\pm1.05}$ | $64.93_{\pm0.00}$ | $74.29_{\pm1.86}$ | $52.31_{\pm0.06}$ | $53.65_{\pm0.24}$ | $53.65_{\pm0.14}$ |
| | M | $84.18_{\pm1.00}$ | $84.44_{\pm0.88}$ | $82.05_{\pm0.72}$ | $79.33_{\pm1.24}$ | $84.11_{\pm0.34}$ | $76.68_{\pm1.05}$ | $73.88_{\pm1.64}$ | $72.72_{\pm1.68}$ | $63.06_{\pm0.00}$ | $80.11_{\pm1.08}$ | $48.89_{\pm0.05}$ | $51.70_{\pm0.32}$ | $51.57_{\pm0.11}$ |
| | H | $88.99_{\pm0.51}$ | $90.71_{\pm0.31}$ | $90.19_{\pm0.41}$ | $87.20_{\pm0.75}$ | $90.04_{\pm0.24}$ | $84.52_{\pm0.73}$ | $80.03_{\pm1.11}$ | $82.87_{\pm0.83}$ | $69.03_{\pm0.00}$ | $89.29_{\pm0.75}$ | $44.74_{\pm0.07}$ | $49.19_{\pm0.26}$ | $48.77_{\pm0.18}$ |
| | F | $85.36_{\pm0.41}$ | $84.95_{\pm0.58}$ | $82.85_{\pm1.29}$ | $79.53_{\pm0.74}$ | $84.22_{\pm0.58}$ | $76.75_{\pm0.89}$ | $68.93_{\pm0.92}$ | $74.11_{\pm0.71}$ | $65.67_{\pm0.00}$ | $81.34_{\pm0.60}$ | $48.32_{\pm0.04}$ | $50.98_{\pm0.17}$ | $50.74_{\pm0.07}$ |
| 6 W/O Attack | E | $84.70_{\pm0.00}$ | $81.34_{\pm0.00}$ | $81.72_{\pm0.00}$ | $82.09_{\pm0.00}$ | $79.10_{\pm0.00}$ | $70.15_{\pm0.00}$ | $77.99_{\pm0.00}$ | $68.28_{\pm0.00}$ | $64.93_{\pm0.00}$ | $79.10_{\pm0.00}$ | $52.04_{\pm0.01}$ | $53.70_{\pm0.08}$ | $54.15_{\pm0.04}$ |
| | M | $83.96_{\pm0.00}$ | $84.33_{\pm0.00}$ | $82.84_{\pm0.00}$ | $83.21_{\pm0.00}$ | $85.45_{\pm0.00}$ | $76.87_{\pm0.00}$ | $82.46_{\pm0.00}$ | $73.51_{\pm0.00}$ | $63.06_{\pm0.00}$ | $81.72_{\pm0.00}$ | $49.67_{\pm0.01}$ | $52.67_{\pm0.50}$ | $52.65_{\pm0.09}$ |
| | H | $89.55_{\pm0.00}$ | $91.04_{\pm0.00}$ | $91.04_{\pm0.00}$ | $89.18_{\pm0.00}$ | $90.67_{\pm0.00}$ | $84.70_{\pm0.00}$ | $88.06_{\pm0.00}$ | $82.84_{\pm0.00}$ | $69.03_{\pm0.00}$ | $89.93_{\pm0.00}$ | $45.94_{\pm0.01}$ | $50.01_{\pm0.56}$ | $49.99_{\pm0.00}$ |
| | F | $86.07_{\pm0.00}$ | $85.57_{\pm0.00}$ | $85.20_{\pm0.00}$ | $84.83_{\pm0.00}$ | $85.07_{\pm0.00}$ | $77.24_{\pm0.00}$ | $82.84_{\pm0.00}$ | $74.88_{\pm0.00}$ | $65.67_{\pm0.00}$ | $83.58_{\pm0.00}$ | $49.21_{\pm0.01}$ | $51.82_{\pm0.33}$ | $51.77_{\pm0.04}$ |
| Avg. Accuracy | E | $82.33_{\pm0.40}$ | $80.65_{\pm0.54}$ | $74.05_{\pm1.05}$ | $72.20_{\pm1.01}$ | $73.82_{\pm0.78}$ | $67.31_{\pm0.56}$ | $68.76_{\pm0.54}$ | $63.10_{\pm0.38}$ | $64.93_{\pm0.00}$ | $67.39_{\pm0.84}$ | - | - | - |
| | M | $84.05_{\pm0.34}$ | $84.58_{\pm0.21}$ | $81.94_{\pm0.30}$ | $79.61_{\pm0.60}$ | $82.33_{\pm0.49}$ | $74.56_{\pm0.45}$ | $75.46_{\pm0.50}$ | $69.80_{\pm0.92}$ | $63.06_{\pm0.00}$ | $77.27_{\pm0.49}$ | - | - | - |
| | H | $88.89_{\pm0.41}$ | $90.49_{\pm0.18}$ | $90.02_{\pm0.27}$ | $87.29_{\pm0.33}$ | $89.50_{\pm0.41}$ | $84.36_{\pm0.36}$ | $80.77_{\pm0.23}$ | $78.60_{\pm0.96}$ | $69.03_{\pm0.00}$ | $88.31_{\pm0.08}$ | - | - | - |
| | F | $85.28_{\pm0.15}$ | $85.20_{\pm0.21}$ | $80.75_{\pm0.41}$ | $78.69_{\pm0.27}$ | $81.15_{\pm0.42}$ | $75.29_{\pm0.49}$ | $71.10_{\pm0.40}$ | $70.91_{\pm0.43}$ | $65.67_{\pm0.00}$ | $75.46_{\pm0.49}$ | - | - | - |
| Avg. 3-Min Accuracy | E | $80.93_{\pm0.61}$ | $79.91_{\pm0.61}$ | $70.14_{\pm1.74}$ | $67.61_{\pm1.32}$ | $69.58_{\pm1.09}$ | $63.92_{\pm0.83}$ | $64.89_{\pm0.88}$ | $59.17_{\pm0.66}$ | $64.93_{\pm0.00}$ | $60.34_{\pm1.48}$ | - | - | - |
| | M | $83.46_{\pm0.31}$ | $84.14_{\pm0.22}$ | $81.07_{\pm0.36}$ | $77.74_{\pm0.93}$ | $79.90_{\pm0.82}$ | $71.29_{\pm1.07}$ | $73.07_{\pm0.83}$ | $67.10_{\pm1.28}$ | $63.06_{\pm0.00}$ | $74.28_{\pm0.90}$ | - | - | - |
| | H | $88.36_{\pm0.28}$ | $90.06_{\pm0.34}$ | $89.39_{\pm0.37}$ | $86.17_{\pm0.57}$ | $88.58_{\pm0.80}$ | $82.97_{\pm0.42}$ | $78.41_{\pm0.38}$ | $74.66_{\pm1.77}$ | $69.03_{\pm0.00}$ | $87.16_{\pm0.61}$ | - | - | - |
| | F | $84.79_{\pm0.23}$ | $84.77_{\pm0.34}$ | $78.01_{\pm0.50}$ | $75.99_{\pm0.42}$ | $78.07_{\pm0.78}$ | $72.46_{\pm0.81}$ | $67.10_{\pm0.76}$ | $68.12_{\pm0.65}$ | $65.67_{\pm0.00}$ | $69.83_{\pm0.88}$ | - | - | - |
| Weighted Accuracy | E | $80.19_{\pm0.86}$ | $79.62_{\pm0.61}$ | $68.40_{\pm1.64}$ | $66.85_{\pm1.66}$ | $66.67_{\pm1.12}$ | $59.85_{\pm1.37}$ | $64.50_{\pm0.99}$ | $57.05_{\pm1.00}$ | $64.93_{\pm0.00}$ | $54.51_{\pm1.73}$ | - | - | - |
| | M | $83.27_{\pm0.40}$ | $84.00_{\pm0.30}$ | $80.70_{\pm0.44}$ | $77.37_{\pm1.24}$ | $77.02_{\pm1.13}$ | $67.76_{\pm0.85}$ | $72.63_{\pm1.04}$ | $65.40_{\pm1.27}$ | $63.06_{\pm0.00}$ | $71.83_{\pm1.38}$ | - | - | - |
| | H | $88.22_{\pm0.31}$ | $89.80_{\pm0.60}$ | $89.15_{\pm0.49}$ | $85.91_{\pm0.56}$ | $88.05_{\pm1.28}$ | $81.70_{\pm0.69}$ | $78.00_{\pm0.67}$ | $69.13_{\pm2.45}$ | $69.03_{\pm0.00}$ | $86.71_{\pm0.94}$ | - | - | - |
| | F | $84.65_{\pm0.23}$ | $84.62_{\pm0.33}$ | $76.86_{\pm0.58}$ | $75.45_{\pm0.96}$ | $74.96_{\pm0.91}$ | $68.64_{\pm0.68}$ | $66.79_{\pm0.98}$ | $65.73_{\pm0.89}$ | $65.67_{\pm0.00}$ | $64.82_{\pm0.60}$ | - | - | - |

Table 14: *grb-citeseer* leaderboard (Top 5 ATK. vs. Top 10 DEF.) in *graph injection* scenario.

| Attack | | 1 GAT+AT | 2 R-GCN+AT | 3 SAGE_AT | 4 GATGuard | 5 GCNGuard | 6 SGCN+LN | 7 R-GCN | 8 GIN+LN | 9 TAGCN+LN | 10 GIN+AT | Avg. Accuracy | Avg. 3-Max Accuracy | Weighted Accuracy |
|---|---|---|---|---|---|---|---|---|---|---|---|---|---|---|
| **1 SPEIT** | E | 70.88±0.83 | 67.49±1.01 | 68.68±1.64 | 65.52±0.00 | 56.43±0.00 | 49.78±2.57 | 52.29±2.76 | 48.53±1.60 | 44.89±4.98 | 47.30±8.94 | 52.01±0.09 | 53.48±0.25 | 53.44±0.17 |
| | M | 72.16±0.44 | 71.00±1.12 | 70.81±1.79 | 63.64±0.00 | 62.70±0.00 | 55.05±1.74 | 61.63±3.24 | 49.78±2.12 | 52.66±4.65 | 47.34±9.06 | 47.31±0.09 | 50.35±1.25 | 50.89±0.24 |
| | H | 78.18±0.49 | 79.69±0.37 | 77.46±0.67 | 72.10±0.00 | 74.64±0.09 | 66.87±1.73 | 77.08±1.91 | 56.86±1.31 | 69.03±4.80 | 51.10±6.38 | 42.72±0.07 | 47.34±2.27 | 48.44±0.12 |
| | F | 73.85±0.22 | 71.30±0.64 | 71.62±1.39 | 67.08±0.00 | 64.54±0.13 | 53.64±1.43 | 59.32±3.65 | 52.82±0.93 | 51.09±6.38 | 44.71±9.62 | 45.86±0.05 | 48.56±0.98 | 48.95±0.12 |
| **2 TDGIA** | E | 71.00±0.67 | 66.68±1.30 | 68.18±1.33 | 65.52±0.00 | 56.43±0.00 | 57.15±1.87 | 48.81±3.56 | 53.92±2.98 | 47.27±2.58 | 51.95±10.85 | 50.12±0.14 | 53.22±1.01 | 53.30±0.12 |
| | M | 72.54±0.47 | 71.76±1.21 | 71.13±1.10 | 63.64±0.00 | 62.70±0.00 | 60.91±0.95 | 60.97±2.82 | 49.22±3.91 | 49.09±6.61 | 51.69±7.68 | 47.21±0.06 | 51.58±0.22 | 51.06±0.10 |
| | H | 78.28±0.37 | 79.72±0.49 | 77.27±0.87 | 72.10±0.00 | 74.61±0.00 | 70.50±1.89 | 73.92±1.91 | 57.80±4.13 | 65.61±3.54 | 66.08±6.43 | 45.17±0.07 | 48.53±1.14 | 48.68±0.06 |
| | F | 73.89±0.40 | 71.75±0.78 | 72.25±0.62 | 67.08±0.00 | 64.58±0.00 | 59.76±2.10 | 56.69±1.77 | 52.16±2.47 | 46.87±4.99 | 59.01±5.47 | 46.24±0.04 | 49.75±0.84 | 49.73±0.09 |
| **3 FGSM** | E | 71.44±1.03 | 69.00±0.75 | 69.28±0.73 | 65.52±0.00 | 56.46±0.17 | 52.38±1.56 | 49.87±2.20 | 52.38±2.41 | 50.88±4.01 | 56.61±2.45 | 50.11±0.11 | 53.19±0.98 | 53.29±0.14 |
| | M | 72.47±0.48 | 73.23±0.51 | 71.25±0.67 | 63.64±0.00 | 62.67±0.10 | 60.22±1.19 | 58.53±1.76 | 57.43±1.53 | 61.76±2.49 | 61.85±1.49 | 47.24±0.08 | 51.64±0.20 | 51.11±0.09 |
| | H | 78.40±0.38 | 79.94±0.31 | 77.24±0.96 | 72.10±0.00 | 74.36±0.75 | 71.03±1.09 | 71.94±1.26 | 70.72±1.29 | 77.90±1.40 | 71.66±2.18 | 45.18±0.05 | 48.50±1.14 | 48.68±0.06 |
| | F | 73.86±0.28 | 73.63±0.50 | 72.40±0.60 | 67.08±0.00 | 64.56±0.04 | 58.05±0.88 | 55.07±1.43 | 61.36±1.03 | 63.39±1.21 | 62.54±0.81 | 46.26±0.04 | 49.81±0.89 | 49.83±0.08 |
| **4 PGD** | E | 71.22±0.75 | 69.19±0.66 | 69.06±0.79 | 65.52±0.00 | 56.40±0.10 | 53.39±1.94 | 47.77±1.29 | 54.70±1.99 | 51.16±2.93 | 58.02±2.31 | 52.72±0.03 | 54.71±0.33 | 54.71±0.07 |
| | M | 72.60±0.67 | 72.91±0.61 | 70.91±0.64 | 63.64±0.00 | 62.70±0.00 | 60.34±1.00 | 57.77±1.58 | 58.78±1.70 | 62.10±1.96 | 60.75±3.33 | 48.71±0.16 | 51.81±0.67 | 51.93±0.09 |
| | H | 78.18±0.40 | 79.94±0.34 | 77.53±0.93 | 72.10±0.00 | 74.61±0.00 | 70.69±1.56 | 71.79±1.67 | 71.57±1.84 | 78.21±0.97 | 71.41±1.80 | 43.58±0.08 | 48.20±1.74 | 48.84±0.08 |
| | F | 73.84±0.26 | 73.58±0.36 | 72.38±0.54 | 67.08±0.00 | 64.46±0.16 | 58.31±0.62 | 54.90±1.58 | 61.60±0.93 | 64.25±1.48 | 63.21±1.19 | 48.26±0.03 | 51.45±0.77 | 51.61±0.06 |
| **5 RND** | E | 71.07±0.47 | 67.56±1.04 | 68.34±0.54 | 65.52±0.00 | 56.34±0.28 | 54.64±2.04 | 51.94±1.64 | 60.88±1.22 | 69.06±1.48 | 60.66±1.42 | 52.31±0.06 | 53.65±0.24 | 53.65±0.14 |
| | M | 72.48±0.34 | 71.82±0.41 | 71.13±0.57 | 63.64±0.00 | 62.73±0.09 | 57.21±1.82 | 61.38±1.08 | 63.10±1.93 | 70.53±1.23 | 62.79±1.02 | 48.89±0.05 | 51.70±0.32 | 51.57±0.11 |
| | H | 78.37±0.44 | 79.84±0.42 | 77.93±0.72 | 72.10±0.00 | 74.58±0.22 | 68.18±1.85 | 75.24±1.81 | 72.41±1.34 | 78.37±0.87 | 72.41±0.40 | 44.74±0.07 | 49.19±0.26 | 48.77±0.18 |
| | F | 74.00±0.45 | 72.95±0.68 | 72.70±0.47 | 67.08±0.00 | 64.55±0.16 | 55.79±0.95 | 60.43±0.71 | 65.66±0.43 | 73.06±0.58 | 65.24±0.51 | 48.32±0.04 | 50.98±0.17 | 50.74±0.07 |
| **6 W/O Attack** | E | 70.22±0.00 | 71.16±0.00 | 70.22±0.00 | 65.52±0.00 | 56.43±0.00 | 71.16±0.00 | 68.75±0.15 | 63.95±0.00 | 69.59±0.00 | 64.26±0.00 | 52.04±0.01 | 53.70±0.88 | 54.15±0.08 |
| | M | 73.04±0.00 | 73.04±0.00 | 71.79±0.00 | 63.64±0.00 | 62.70±0.00 | 75.55±0.00 | 70.16±0.19 | 64.58±0.00 | 73.67±0.00 | 65.20±0.00 | 49.67±0.01 | 52.67±0.50 | 52.65±0.09 |
| | H | 78.37±0.00 | 80.88±0.00 | 78.06±0.00 | 72.10±0.00 | 74.61±0.00 | 79.62±0.00 | 80.41±0.16 | 69.59±0.00 | 77.43±0.00 | 73.98±0.00 | 45.94±0.01 | 50.01±0.56 | 49.99±0.00 |
| | F | 73.88±0.00 | 75.03±0.00 | 73.35±0.00 | 67.08±0.00 | 64.58±0.00 | 75.44±0.00 | 73.16±0.00 | 66.04±0.00 | 73.56±0.00 | 67.82±0.00 | 49.21±0.01 | 51.82±0.33 | 51.77±0.04 |
| **Avg. Accuracy** | E | 70.97±0.26 | 68.51±0.41 | 68.96±0.47 | 65.52±0.00 | 56.41±0.06 | 56.42±0.53 | 53.24±0.73 | 55.73±0.42 | 55.48±1.32 | 56.47±2.73 | - | - | - |
| | M | 72.55±0.21 | 72.29±0.26 | 71.17±0.34 | 63.64±0.00 | 62.70±0.02 | 61.55±0.65 | 61.74±0.93 | 57.15±1.00 | 61.64±1.06 | 58.27±1.47 | - | - | - |
| | H | 78.30±0.17 | 80.00±0.13 | 77.58±0.29 | 72.10±0.00 | 74.57±0.13 | 71.15±0.50 | 75.06±0.71 | 66.49±0.67 | 74.42±1.31 | 67.78±1.47 | - | - | - |
| | F | 73.89±0.11 | 73.04±0.17 | 72.45±0.17 | 67.08±0.00 | 64.54±0.04 | 60.16±0.55 | 59.93±0.57 | 59.94±0.63 | 62.04±1.70 | 60.42±1.30 | - | - | - |
| **Avg. 3-Min Accuracy** | E | 70.37±0.29 | 67.19±0.80 | 68.08±0.64 | 65.52±0.00 | 56.38±0.10 | 51.58±0.94 | 48.25±0.95 | 51.09±0.74 | 47.07±1.85 | 51.32±4.97 | - | - | - |
| | M | 72.17±0.26 | 71.44±0.37 | 70.50±0.56 | 63.64±0.00 | 62.69±0.03 | 57.22±1.16 | 58.43±1.16 | 51.98±1.52 | 54.06±2.00 | 52.96±3.03 | - | - | - |
| | H | 78.05±0.20 | 79.59±0.21 | 77.00±0.40 | 72.10±0.00 | 74.51±0.25 | 68.16±0.90 | 72.31±0.96 | 61.27±1.55 | 70.49±2.41 | 62.42±2.81 | - | - | - |
| | F | 73.68±0.12 | 71.94±0.36 | 71.86±0.30 | 67.08±0.00 | 64.49±0.08 | 55.61±0.58 | 55.19±0.62 | 55.26±1.07 | 53.69±3.32 | 55.31±2.84 | - | - | - |
| **Weighted Accuracy** | E | 70.26±0.29 | 66.91±0.80 | 67.73±0.82 | 65.52±0.00 | 56.33±0.19 | 50.83±1.23 | 47.83±1.09 | 49.98±0.93 | 46.26±2.89 | 47.56±6.43 | - | - | - |
| | M | 72.04±0.27 | 71.03±0.58 | 70.09±0.74 | 63.64±0.00 | 62.68±0.06 | 56.51±1.19 | 57.80±1.03 | 50.29±2.25 | 51.08±3.58 | 48.58±5.29 | - | - | - |
| | H | 77.94±0.22 | 79.53±0.22 | 76.79±0.41 | 72.10±0.00 | 74.40±0.51 | 67.47±1.09 | 71.97±1.10 | 58.78±1.41 | 67.53±2.85 | 56.76±4.16 | - | - | - |
| | F | 73.64±0.15 | 71.59±0.34 | 71.48±0.66 | 67.08±0.00 | 64.44±0.12 | 55.05±1.00 | 54.95±0.76 | 53.66±1.23 | 49.94±3.72 | 49.84±6.03 | - | - | - |

Table 15: *grb-flickr* leaderboard (Top 5 ATK. vs. Top 10 DEF.) in *graph injection* scenario.

| Attack | | 1 R-GCN+AT | 2 GAT+LN | 3 SAGE+LN | 4 GIN_LN | 5 GCN+AT | 6 SAGE+AT | 7 GAT | 8 GIN+AT | 9 SAGE | 10 APPNP+LN | Avg. Accuracy | Avg. 3-Max Accuracy | Weighted Accuracy |
|---|---|---|---|---|---|---|---|---|---|---|---|---|---|---|
| **1 SPEIT** | E | 53.41±0.12 | 53.36±0.28 | 51.43±0.15 | 50.96±0.29 | 52.89±0.22 | 52.87±0.13 | 53.62±0.36 | 50.15±0.14 | 51.65±0.45 | 49.78±0.03 | 52.01±0.09 | 53.48±0.25 | 53.44±0.17 |
| | M | 49.79±0.26 | 49.21±0.26 | 47.83±0.09 | 46.19±0.14 | 48.05±0.24 | 48.91±0.12 | 52.04±0.33 | 43.62±0.10 | 44.54±0.12 | 42.94±0.03 | 47.31±0.09 | 50.35±1.25 | 50.89±0.24 |
| | H | 44.72±0.15 | 46.78±0.54 | 43.41±0.12 | 44.79±0.09 | 44.83±0.14 | 43.51±0.15 | 50.32±0.20 | 37.33±0.12 | 37.17±0.17 | 34.33±0.02 | 42.72±0.07 | 47.34±2.27 | 48.44±0.12 |
| | F | 47.14±0.13 | 46.39±0.13 | 47.62±0.04 | 46.18±0.09 | 43.52±0.07 | 48.17±0.14 | 49.91±0.17 | 43.73±0.05 | 43.66±0.07 | 42.30±0.01 | 45.86±0.05 | 48.56±0.98 | 48.95±0.12 |
| **2 FGSM** | E | 53.99±0.13 | 51.87±0.49 | 50.89±0.15 | 48.24±0.15 | 53.79±0.17 | 44.06±0.23 | 48.23±0.56 | 50.08±0.16 | 49.99±0.33 | 50.07±0.05 | 50.12±0.14 | 53.22±1.01 | 53.30±0.12 |
| | M | 51.79±0.13 | 51.42±0.20 | 47.89±0.13 | 46.43±0.14 | 51.54±0.13 | 44.24±0.17 | 46.17±0.41 | 44.81±0.04 | 44.24±0.22 | 43.59±0.06 | 47.21±0.06 | 51.58±0.22 | 51.06±0.10 |
| | H | 46.87±0.12 | 49.32±0.09 | 43.69±0.11 | 46.76±0.16 | 49.35±0.08 | 46.46±0.22 | 45.24±0.29 | 42.58±0.30 | 45.90±0.16 | 35.56±0.08 | 45.17±0.07 | 48.53±1.14 | 48.68±0.06 |
| | F | 50.20±0.08 | 50.47±0.15 | 47.58±0.08 | 46.94±0.07 | 48.58±0.09 | 43.10±0.14 | 42.78±0.27 | 45.28±0.10 | 43.33±0.19 | 44.16±0.09 | 46.24±0.04 | 49.75±0.84 | 49.73±0.09 |
| **3 PGD** | E | 54.02±0.18 | 51.88±0.40 | 50.77±0.19 | 48.36±0.19 | 53.68±0.18 | 43.95±0.24 | 48.33±0.71 | 50.04±0.15 | 49.95±0.42 | 50.05±0.07 | 50.11±0.11 | 53.19±0.98 | 53.29±0.14 |
| | M | 51.74±0.22 | 51.65±0.21 | 47.77±0.16 | 46.69±0.23 | 51.53±0.10 | 44.26±0.17 | 45.91±0.40 | 44.95±0.27 | 44.38±0.43 | 43.56±0.10 | 47.24±0.08 | 51.64±0.20 | 51.11±0.09 |
| | H | 46.81±0.13 | 49.28±0.13 | 43.75±0.17 | 46.75±0.22 | 49.32±0.14 | 46.45±0.18 | 45.24±0.26 | 42.47±0.27 | 46.17±0.19 | 35.55±0.08 | 45.18±0.05 | 48.50±1.14 | 48.68±0.06 |
| | F | 50.22±0.11 | 51.66±0.08 | 47.56±0.07 | 46.94±0.10 | 48.58±0.09 | 43.09±0.15 | 42.90±0.24 | 45.24±0.10 | 43.32±0.14 | 44.14±0.07 | 46.26±0.04 | 49.81±0.89 | 49.83±0.08 |
| **4 TDGIA** | E | 55.13±0.08 | 54.17±0.17 | 51.83±0.08 | 52.29±0.06 | 54.35±0.07 | 53.61±0.05 | 54.66±0.12 | 49.95±0.02 | 51.50±0.12 | 49.72±0.01 | 52.72±0.03 | 54.71±0.33 | 54.71±0.07 |
| | M | 51.45±0.12 | 51.30±0.44 | 46.86±0.08 | 48.54±0.17 | 50.86±0.31 | 50.03±0.16 | 52.67±0.24 | 43.20±0.07 | 49.31±1.04 | 42.88±0.00 | 48.71±0.16 | 51.81±0.67 | 51.93±0.09 |
| | H | 45.20±0.10 | 48.19±0.17 | 42.04±0.09 | 46.09±0.07 | 44.97±0.30 | 43.16±0.09 | 50.33±0.13 | 37.33±0.05 | 44.04±0.08 | 34.41±0.04 | 43.58±0.08 | 48.20±1.74 | 48.84±0.08 |
| | F | 48.97±0.05 | 51.25±0.07 | 47.13±0.07 | 49.05±0.05 | 47.54±0.06 | 49.52±0.05 | 52.46±0.09 | 43.77±0.01 | 50.63±0.08 | 42.30±0.00 | 48.26±0.03 | 51.45±0.06 | 51.61±0.06 |
| **5 RND** | E | 53.90±0.30 | 53.31±0.18 | 51.73±0.14 | 51.16±0.15 | 53.62±0.10 | 53.19±0.12 | 52.77±0.22 | 50.36±0.08 | 53.25±0.23 | 49.83±0.03 | 52.31±0.06 | 53.65±0.24 | 53.65±0.14 |
| | M | 51.22±0.15 | 52.01±0.15 | 47.82±0.12 | 48.21±0.15 | 51.78±0.14 | 49.84±0.19 | 51.06±0.30 | 44.11±0.03 | 49.85±0.18 | 43.04±0.06 | 48.89±0.05 | 51.70±0.32 | 51.57±0.11 |
| | H | 46.38±0.15 | 49.33±0.18 | 42.90±0.13 | 46.79±0.14 | 48.95±0.12 | 45.03±0.15 | 49.29±0.25 | 39.10±0.09 | 45.13±0.28 | 34.50±0.03 | 44.74±0.07 | 49.19±0.26 | 48.77±0.18 |
| | F | 49.32±0.10 | 51.09±0.08 | 47.47±0.07 | 48.58±0.00 | 51.00±0.09 | 49.18±0.07 | 50.84±0.20 | 44.36±0.03 | 49.00±0.08 | 42.31±0.01 | 48.32±0.04 | 50.98±0.17 | 50.74±0.07 |
| **6 W/O Attack** | E | 54.94±0.12 | 52.58±0.00 | 51.68±0.00 | 50.48±0.00 | 52.97±0.00 | 53.18±0.00 | 49.50±0.00 | 50.72±0.00 | 52.77±0.00 | 51.55±0.00 | 52.04±0.01 | 53.70±0.88 | 54.15±0.08 |
| | M | 53.23±0.13 | 52.75±0.00 | 47.83±0.00 | 48.03±0.00 | 52.03±0.00 | 51.27±0.00 | 50.06±0.00 | 44.87±0.00 | 51.63±0.00 | 45.01±0.00 | 49.67±0.01 | 52.67±0.50 | 52.65±0.09 |
| | H | 48.70±0.08 | 49.65±0.00 | 42.90±0.00 | 46.43±0.00 | 49.59±0.00 | 47.10±0.00 | 50.80±0.00 | 40.40±0.00 | 47.96±0.00 | 36.12±0.00 | 45.94±0.01 | 50.01±0.56 | 49.99±0.00 |
| | F | 52.28±0.06 | 51.66±0.00 | 47.47±0.00 | 48.31±0.00 | 51.53±0.00 | 50.52±0.00 | 50.12±0.00 | 45.23±0.00 | 50.79±0.00 | 44.23±0.00 | 49.21±0.01 | 51.82±0.33 | 51.77±0.04 |
| **Avg. Accuracy** | E | 54.23±0.07 | 52.86±0.16 | 51.38±0.05 | 50.25±0.09 | 53.55±0.07 | 50.14±0.07 | 51.19±0.15 | 50.22±0.04 | 51.52±0.12 | 50.17±0.02 | - | - | - |
| | M | 51.54±0.07 | 51.39±0.10 | 47.67±0.05 | 47.35±0.06 | 50.96±0.04 | 48.09±0.07 | 49.65±0.09 | 44.26±0.06 | 47.32±0.19 | 43.50±0.02 | - | - | - |
| | H | 46.46±0.05 | 48.76±0.11 | 43.12±0.05 | 46.27±0.04 | 47.84±0.08 | 45.29±0.04 | 48.54±0.08 | 39.82±0.06 | 44.40±0.10 | 35.08±0.02 | - | - | - |
| | F | 49.69±0.04 | 50.25±0.03 | 47.47±0.03 | 47.67±0.03 | 48.46±0.02 | 47.26±0.04 | 48.17±0.09 | 44.60±0.03 | 46.79±0.06 | 43.24±0.01 | - | - | - |
| **Avg. 3-Min Accuracy** | E | 53.74±0.09 | 52.11±0.26 | 51.03±0.10 | 49.03±0.07 | 53.14±0.09 | 46.96±0.14 | 48.69±0.25 | 50.00±0.07 | 50.46±0.22 | 49.78±0.02 | - | - | - |
| | M | 50.81±0.10 | 50.62±0.17 | 47.44±0.07 | 46.44±0.11 | 50.13±0.09 | 45.81±0.09 | 47.38±0.19 | 43.65±0.04 | 44.38±0.21 | 42.95±0.02 | - | - | - |
| | H | 45.45±0.07 | 48.05±0.21 | 42.62±0.05 | 45.77±0.04 | 46.25±0.15 | 43.90±0.08 | 46.59±0.17 | 37.92±0.04 | 42.11±0.15 | 34.41±0.01 | - | - | - |
| | F | 48.48±0.06 | 49.16±0.04 | 47.35±0.04 | 46.69±0.04 | 46.52±0.04 | 44.79±0.07 | 45.19±0.14 | 43.95±0.02 | 43.44±0.10 | 42.30±0.00 | - | - | - |
| **Weighted Accuracy** | E | 53.62±0.08 | 51.97±0.33 | 50.92±0.12 | 48.67±0.09 | 53.02±0.12 | 45.43±0.18 | 48.62±0.42 | 49.97±0.06 | 50.22±0.26 | 49.80±0.01 | - | - | - |
| | M | 50.35±0.14 | 49.95±0.18 | 47.16±0.07 | 46.44±0.10 | 49.10±0.14 | 45.06±0.09 | 46.70±0.28 | 43.48±0.04 | 44.73±0.21 | 42.99±0.01 | - | - | - |
| | H | 45.15±0.10 | 47.43±0.37 | 42.37±0.06 | 45.30±0.07 | 45.55±0.13 | 43.66±0.07 | 45.92±0.18 | 37.78±0.04 | 39.72±0.13 | 34.47±0.01 | - | - | - |
| | F | 47.91±0.09 | 47.81±0.08 | 47.25±0.05 | 46.57±0.06 | 45.13±0.05 | 43.99±0.09 | 43.98±0.18 | 43.91±0.03 | 43.88±0.10 | 42.46±0.01 | - | - | - |

Table 16: *grb-reddit* leaderboard (Top 5 ATK. vs. Top 10 DEF.) in *graph injection* scenario.

| Attack | | 1 GIN+LN | 2 TAGCN+LN | 3 TAGCN+AT | 4 GAT+LN | 5 R-GCN+AT | 6 TAGCN | 7 GCN+LN | 8 SAGE+AT | 9 SAGE | 10 SGCN+LN | Avg. Accuracy | Avg. 3-Max Accuracy | Weighted Accuracy |
|---|---|---|---|---|---|---|---|---|---|---|---|---|---|---|
| **1 TDGIA** | E | $89.57_{\pm.05}$ | $90.59_{\pm.02}$ | $70.17_{\pm.28}$ | $84.18_{\pm.03}$ | $88.38_{\pm.02}$ | $78.83_{\pm.18}$ | $80.79_{\pm.17}$ | $86.29_{\pm.03}$ | $71.00_{\pm.11}$ | $76.36_{\pm.03}$ | $81.62_{\pm.08}$ | $89.52_{\pm.90}$ | $89.17_{\pm.03}$ |
| | M | $98.29_{\pm.01}$ | $97.79_{\pm.00}$ | $98.02_{\pm.01}$ | $96.20_{\pm.01}$ | $95.50_{\pm.01}$ | $96.62_{\pm.01}$ | $97.66_{\pm.00}$ | $94.07_{\pm.01}$ | $95.66_{\pm.02}$ | $92.27_{\pm.01}$ | $96.21_{\pm.00}$ | $98.03_{\pm.21}$ | $97.97_{\pm.01}$ |
| | H | $99.54_{\pm.00}$ | $99.16_{\pm.00}$ | $98.54_{\pm.01}$ | $98.17_{\pm.01}$ | $95.68_{\pm.02}$ | $97.77_{\pm.01}$ | $99.14_{\pm.01}$ | $90.60_{\pm.01}$ | $98.43_{\pm.02}$ | $93.33_{\pm.01}$ | $97.04_{\pm.01}$ | $99.28_{\pm.18}$ | $99.18_{\pm.00}$ |
| | F | $95.92_{\pm.01}$ | $95.89_{\pm.01}$ | $93.12_{\pm.00}$ | $93.73_{\pm.01}$ | $93.08_{\pm.01}$ | $91.77_{\pm.01}$ | $91.09_{\pm.02}$ | $90.16_{\pm.01}$ | $85.98_{\pm.03}$ | $86.61_{\pm.01}$ | $91.74_{\pm.00}$ | $95.18_{\pm1.03}$ | $95.24_{\pm.01}$ |
| **2 SPEIT** | E | $91.92_{\pm.04}$ | $91.58_{\pm.03}$ | $91.73_{\pm.08}$ | $87.18_{\pm.08}$ | $88.81_{\pm.02}$ | $88.41_{\pm.05}$ | $89.81_{\pm.05}$ | $86.65_{\pm.03}$ | $83.10_{\pm.07}$ | $76.55_{\pm.10}$ | $87.57_{\pm.02}$ | $91.74_{\pm.14}$ | $91.35_{\pm.03}$ |
| | M | $98.27_{\pm.01}$ | $97.84_{\pm.01}$ | $97.98_{\pm.02}$ | $96.19_{\pm.02}$ | $95.28_{\pm.01}$ | $96.60_{\pm.02}$ | $97.77_{\pm.01}$ | $94.03_{\pm.02}$ | $96.74_{\pm.02}$ | $92.26_{\pm.03}$ | $96.30_{\pm.00}$ | $98.03_{\pm.18}$ | $97.97_{\pm.01}$ |
| | H | $99.54_{\pm.00}$ | $99.20_{\pm.01}$ | $98.81_{\pm.00}$ | $98.17_{\pm.01}$ | $96.04_{\pm.01}$ | $97.97_{\pm.01}$ | $99.22_{\pm.01}$ | $90.69_{\pm.03}$ | $98.22_{\pm.03}$ | $93.43_{\pm.02}$ | $97.13_{\pm.00}$ | $99.32_{\pm.16}$ | $99.21_{\pm.01}$ |
| | F | $96.31_{\pm.02}$ | $96.31_{\pm.02}$ | $95.77_{\pm.02}$ | $93.76_{\pm.02}$ | $93.59_{\pm.01}$ | $93.76_{\pm.03}$ | $95.57_{\pm.01}$ | $90.21_{\pm.02}$ | $92.96_{\pm.04}$ | $87.02_{\pm.03}$ | $93.53_{\pm.01}$ | $96.13_{\pm.26}$ | $95.96_{\pm.01}$ |
| **3 FGSM** | E | $91.77_{\pm.06}$ | $91.53_{\pm.05}$ | $92.60_{\pm.04}$ | $87.38_{\pm.05}$ | $89.06_{\pm.01}$ | $90.52_{\pm.05}$ | $89.70_{\pm.03}$ | $86.78_{\pm.03}$ | $88.24_{\pm.06}$ | $78.51_{\pm.12}$ | $88.61_{\pm.02}$ | $91.97_{\pm.46}$ | $91.92_{\pm.03}$ |
| | M | $98.26_{\pm.01}$ | $97.74_{\pm.01}$ | $98.13_{\pm.01}$ | $96.18_{\pm.01}$ | $95.47_{\pm.01}$ | $96.94_{\pm.03}$ | $97.68_{\pm.01}$ | $94.01_{\pm.02}$ | $96.52_{\pm.04}$ | $92.43_{\pm.03}$ | $96.36_{\pm.01}$ | $98.04_{\pm.22}$ | $97.99_{\pm.01}$ |
| | H | $99.55_{\pm.01}$ | $99.02_{\pm.01}$ | $99.16_{\pm.01}$ | $98.17_{\pm.02}$ | $95.94_{\pm.01}$ | $98.32_{\pm.01}$ | $99.08_{\pm.01}$ | $91.01_{\pm.02}$ | $98.48_{\pm.02}$ | $93.65_{\pm.02}$ | $97.24_{\pm.00}$ | $99.27_{\pm.21}$ | $99.23_{\pm.01}$ |
| | F | $96.48_{\pm.01}$ | $95.90_{\pm.01}$ | $96.74_{\pm.01}$ | $93.91_{\pm.01}$ | $93.57_{\pm.01}$ | $95.24_{\pm.02}$ | $95.22_{\pm.01}$ | $90.54_{\pm.02}$ | $93.78_{\pm.03}$ | $89.01_{\pm.03}$ | $94.04_{\pm.01}$ | $96.37_{\pm.35}$ | $96.32_{\pm.01}$ |
| **4 RND** | E | $92.04_{\pm.04}$ | $91.75_{\pm.04}$ | $92.60_{\pm.00}$ | $87.09_{\pm.04}$ | $88.87_{\pm.03}$ | $89.55_{\pm.05}$ | $90.00_{\pm.04}$ | $86.71_{\pm.03}$ | $88.12_{\pm.09}$ | $77.27_{\pm.10}$ | $88.40_{\pm.02}$ | $92.13_{\pm.36}$ | $91.94_{\pm.02}$ |
| | M | $98.28_{\pm.02}$ | $97.82_{\pm.01}$ | $98.13_{\pm.01}$ | $96.17_{\pm.02}$ | $95.42_{\pm.02}$ | $96.84_{\pm.03}$ | $97.75_{\pm.01}$ | $94.07_{\pm.02}$ | $97.14_{\pm.02}$ | $92.43_{\pm.03}$ | $96.40_{\pm.01}$ | $98.08_{\pm.19}$ | $98.02_{\pm.01}$ |
| | H | $99.54_{\pm.01}$ | $99.13_{\pm.01}$ | $99.00_{\pm.01}$ | $98.17_{\pm.01}$ | $96.21_{\pm.02}$ | $98.15_{\pm.01}$ | $99.12_{\pm.00}$ | $90.89_{\pm.03}$ | $98.34_{\pm.03}$ | $93.50_{\pm.01}$ | $97.21_{\pm.01}$ | $99.27_{\pm.19}$ | $99.21_{\pm.00}$ |
| | F | $96.60_{\pm.01}$ | $96.30_{\pm.02}$ | $96.54_{\pm.01}$ | $93.84_{\pm.02}$ | $93.23_{\pm.02}$ | $94.88_{\pm.03}$ | $95.63_{\pm.01}$ | $90.37_{\pm.03}$ | $94.59_{\pm.02}$ | $87.72_{\pm.03}$ | $93.97_{\pm.01}$ | $96.48_{\pm.13}$ | $96.27_{\pm.01}$ |
| **5 PGD** | E | $91.74_{\pm.06}$ | $91.56_{\pm.04}$ | $92.63_{\pm.04}$ | $87.39_{\pm.04}$ | $89.06_{\pm.02}$ | $90.58_{\pm.06}$ | $89.72_{\pm.04}$ | $86.77_{\pm.03}$ | $88.25_{\pm.10}$ | $78.53_{\pm.07}$ | $88.62_{\pm.02}$ | $91.98_{\pm.47}$ | $91.94_{\pm.03}$ |
| | M | $98.26_{\pm.01}$ | $97.74_{\pm.01}$ | $98.12_{\pm.02}$ | $96.19_{\pm.01}$ | $95.46_{\pm.02}$ | $96.94_{\pm.01}$ | $97.68_{\pm.01}$ | $94.01_{\pm.03}$ | $96.52_{\pm.05}$ | $92.45_{\pm.04}$ | $96.36_{\pm.01}$ | $98.04_{\pm.22}$ | $97.98_{\pm.01}$ |
| | H | $99.55_{\pm.01}$ | $99.02_{\pm.01}$ | $99.16_{\pm.01}$ | $98.18_{\pm.01}$ | $95.94_{\pm.01}$ | $98.32_{\pm.01}$ | $99.08_{\pm.01}$ | $91.00_{\pm.03}$ | $98.47_{\pm.03}$ | $93.66_{\pm.02}$ | $97.24_{\pm.00}$ | $99.26_{\pm.21}$ | $99.23_{\pm.00}$ |
| | F | $96.48_{\pm.01}$ | $95.91_{\pm.01}$ | $96.74_{\pm.01}$ | $93.91_{\pm.02}$ | $93.56_{\pm.01}$ | $95.24_{\pm.03}$ | $95.23_{\pm.02}$ | $90.54_{\pm.02}$ | $93.75_{\pm.02}$ | $88.99_{\pm.03}$ | $94.03_{\pm.01}$ | $96.38_{\pm.35}$ | $96.32_{\pm.01}$ |
| **6 W/O Attack** | E | $92.19_{\pm.00}$ | $92.11_{\pm.00}$ | $93.05_{\pm.00}$ | $87.97_{\pm.00}$ | $89.14_{\pm.00}$ | $90.99_{\pm.00}$ | $90.17_{\pm.00}$ | $86.86_{\pm.00}$ | $89.96_{\pm.00}$ | $83.12_{\pm.00}$ | $89.56_{\pm.00}$ | $92.45_{\pm.43}$ | $92.41_{\pm.00}$ |
| | M | $98.30_{\pm.00}$ | $97.82_{\pm.00}$ | $98.23_{\pm.00}$ | $96.27_{\pm.00}$ | $95.75_{\pm.00}$ | $97.07_{\pm.00}$ | $97.74_{\pm.00}$ | $93.86_{\pm.00}$ | $97.21_{\pm.00}$ | $93.48_{\pm.00}$ | $96.57_{\pm.00}$ | $98.12_{\pm.21}$ | $98.06_{\pm.00}$ |
| | H | $99.55_{\pm.00}$ | $99.16_{\pm.00}$ | $99.03_{\pm.00}$ | $98.20_{\pm.00}$ | $96.46_{\pm.00}$ | $98.12_{\pm.00}$ | $99.12_{\pm.00}$ | $90.72_{\pm.00}$ | $98.30_{\pm.00}$ | $93.84_{\pm.00}$ | $97.25_{\pm.00}$ | $99.28_{\pm.19}$ | $99.23_{\pm.00}$ |
| | F | $96.68_{\pm.00}$ | $96.37_{\pm.00}$ | $96.77_{\pm.00}$ | $94.15_{\pm.00}$ | $93.78_{\pm.00}$ | $95.39_{\pm.00}$ | $95.68_{\pm.00}$ | $90.48_{\pm.00}$ | $95.16_{\pm.00}$ | $90.15_{\pm.00}$ | $94.46_{\pm.00}$ | $96.61_{\pm.17}$ | $96.46_{\pm.00}$ |
| **Avg. Accuracy** | E | $91.53_{\pm.01}$ | $91.52_{\pm.01}$ | $88.80_{\pm.05}$ | $86.87_{\pm.02}$ | $88.89_{\pm.01}$ | $88.15_{\pm.03}$ | $88.37_{\pm.04}$ | $86.68_{\pm.01}$ | $84.78_{\pm.03}$ | $78.51_{\pm.04}$ | - | - | - |
| | M | $98.28_{\pm.00}$ | $97.79_{\pm.01}$ | $98.10_{\pm.01}$ | $96.20_{\pm.01}$ | $95.48_{\pm.01}$ | $96.84_{\pm.01}$ | $97.71_{\pm.00}$ | $94.01_{\pm.01}$ | $96.63_{\pm.01}$ | $92.62_{\pm.01}$ | - | - | - |
| | H | $99.54_{\pm.00}$ | $99.12_{\pm.00}$ | $98.95_{\pm.00}$ | $98.17_{\pm.01}$ | $96.05_{\pm.01}$ | $98.11_{\pm.01}$ | $99.13_{\pm.00}$ | $90.82_{\pm.01}$ | $98.37_{\pm.01}$ | $93.57_{\pm.01}$ | - | - | - |
| | F | $96.41_{\pm.01}$ | $96.11_{\pm.01}$ | $95.95_{\pm.01}$ | $93.88_{\pm.01}$ | $93.47_{\pm.01}$ | $94.38_{\pm.01}$ | $94.74_{\pm.01}$ | $90.38_{\pm.01}$ | $92.70_{\pm.01}$ | $88.25_{\pm.01}$ | - | - | - |
| **Avg. 3-Min Accuracy** | E | $91.03_{\pm.02}$ | $91.22_{\pm.02}$ | $84.82_{\pm.10}$ | $86.15_{\pm.03}$ | $88.69_{\pm.01}$ | $85.59_{\pm.07}$ | $86.73_{\pm.07}$ | $86.55_{\pm.01}$ | $80.73_{\pm.05}$ | $76.72_{\pm.05}$ | - | - | - |
| | M | $98.26_{\pm.01}$ | $97.75_{\pm.01}$ | $98.04_{\pm.01}$ | $96.17_{\pm.01}$ | $95.38_{\pm.01}$ | $96.69_{\pm.01}$ | $97.67_{\pm.00}$ | $93.96_{\pm.01}$ | $96.23_{\pm.03}$ | $92.32_{\pm.02}$ | - | - | - |
| | H | $99.54_{\pm.00}$ | $99.06_{\pm.00}$ | $98.78_{\pm.01}$ | $98.16_{\pm.01}$ | $95.86_{\pm.01}$ | $97.95_{\pm.01}$ | $99.09_{\pm.00}$ | $90.67_{\pm.01}$ | $98.28_{\pm.01}$ | $93.42_{\pm.01}$ | - | - | - |
| | F | $96.24_{\pm.01}$ | $95.90_{\pm.01}$ | $95.14_{\pm.02}$ | $93.78_{\pm.01}$ | $93.29_{\pm.01}$ | $93.47_{\pm.01}$ | $93.85_{\pm.01}$ | $90.24_{\pm.02}$ | $90.89_{\pm.02}$ | $87.12_{\pm.01}$ | - | - | - |
| **Weighted Accuracy** | E | $90.31_{\pm.03}$ | $90.92_{\pm.02}$ | $77.42_{\pm.19}$ | $85.18_{\pm.02}$ | $88.55_{\pm.02}$ | $82.26_{\pm.13}$ | $83.75_{\pm.12}$ | $86.43_{\pm.02}$ | $75.84_{\pm.07}$ | $76.73_{\pm.03}$ | - | - | - |
| | M | $98.26_{\pm.01}$ | $97.75_{\pm.01}$ | $98.01_{\pm.01}$ | $96.17_{\pm.01}$ | $95.34_{\pm.01}$ | $96.65_{\pm.01}$ | $97.67_{\pm.00}$ | $93.91_{\pm.00}$ | $95.98_{\pm.02}$ | $92.32_{\pm.02}$ | - | - | - |
| | H | $99.53_{\pm.01}$ | $99.04_{\pm.00}$ | $98.67_{\pm.01}$ | $98.16_{\pm.01}$ | $95.79_{\pm.01}$ | $97.87_{\pm.01}$ | $99.09_{\pm.01}$ | $90.66_{\pm.01}$ | $98.26_{\pm.02}$ | $93.39_{\pm.00}$ | - | - | - |
| | F | $96.09_{\pm.01}$ | $95.93_{\pm.01}$ | $94.14_{\pm.01}$ | $93.76_{\pm.01}$ | $93.19_{\pm.00}$ | $92.64_{\pm.01}$ | $92.49_{\pm.01}$ | $90.22_{\pm.01}$ | $88.46_{\pm.03}$ | $86.99_{\pm.01}$ | - | - | - |

Table 17: *grb-aminer* leaderboard (Top 5 ATK. vs. Top 10 DEF.) in *graph injection* scenario.

| Attacks | | 1 GAT+AT | 2 R-GCN+AT | 3 SGCN+LN | 4 R-GCN | 5 GCN+LN | 6 GAT+LN | 7 GIN+LN | 8 TAGCN+LN | 9 TAGCN+AT | 10 GAT | Avg. Accuracy | Avg. 3-Min Accuracy | Weighted Accuracy |
|---|---|---|---|---|---|---|---|---|---|---|---|---|---|---|
| **1 TDGIA** | E | $59.54_{\pm.08}$ | $56.83_{\pm.06}$ | $56.73_{\pm.06}$ | $56.12_{\pm.07}$ | $53.51_{\pm.21}$ | $43.93_{\pm.41}$ | $51.10_{\pm.12}$ | $54.63_{\pm.20}$ | $49.59_{\pm.50}$ | $42.40_{\pm.52}$ | $52.44_{\pm.17}$ | $57.70_{\pm1.31}$ | $58.08_{\pm.04}$ |
| | M | $68.39_{\pm.02}$ | $65.61_{\pm.02}$ | $66.11_{\pm.02}$ | $65.23_{\pm.03}$ | $66.78_{\pm.05}$ | $61.84_{\pm1.20}$ | $64.49_{\pm.10}$ | $64.62_{\pm.02}$ | $67.27_{\pm.04}$ | $62.47_{\pm.01}$ | $65.28_{\pm.23}$ | $67.48_{\pm.68}$ | $67.69_{\pm.02}$ |
| | H | $75.83_{\pm.02}$ | $72.35_{\pm.02}$ | $72.10_{\pm.00}$ | $71.94_{\pm.02}$ | $73.39_{\pm.02}$ | $75.22_{\pm.04}$ | $72.92_{\pm.02}$ | $68.94_{\pm.03}$ | $73.98_{\pm.01}$ | $75.03_{\pm.03}$ | $73.17_{\pm.01}$ | $75.36_{\pm.34}$ | $75.33_{\pm.01}$ |
| | F | $67.69_{\pm.03}$ | $63.62_{\pm.32}$ | $62.20_{\pm.15}$ | $61.99_{\pm.02}$ | $60.38_{\pm1.46}$ | $59.69_{\pm1.57}$ | $59.59_{\pm.42}$ | $59.06_{\pm1.75}$ | $57.24_{\pm5.04}$ | $56.63_{\pm4.75}$ | $60.81_{\pm1.71}$ | $64.52_{\pm2.32}$ | $65.74_{\pm.21}$ |
| **2 SPEIT** | E | $59.54_{\pm.07}$ | $56.80_{\pm.05}$ | $56.94_{\pm.10}$ | $55.64_{\pm.10}$ | $56.15_{\pm.06}$ | $56.13_{\pm.07}$ | $54.24_{\pm.09}$ | $56.61_{\pm.06}$ | $56.59_{\pm.08}$ | $57.36_{\pm.09}$ | $56.60_{\pm.04}$ | $57.95_{\pm1.14}$ | $58.62_{\pm.05}$ |
| | M | $68.37_{\pm.03}$ | $65.46_{\pm.03}$ | $66.20_{\pm.02}$ | $65.25_{\pm.04}$ | $66.75_{\pm.04}$ | $67.49_{\pm.06}$ | $65.05_{\pm.06}$ | $64.47_{\pm.04}$ | $66.95_{\pm.05}$ | $66.81_{\pm.04}$ | $66.28_{\pm.02}$ | $67.60_{\pm.59}$ | $67.86_{\pm.03}$ |
| | H | $75.94_{\pm.04}$ | $72.27_{\pm.03}$ | $72.36_{\pm.03}$ | $71.86_{\pm.03}$ | $73.41_{\pm.01}$ | $75.34_{\pm.03}$ | $72.87_{\pm.03}$ | $68.88_{\pm.05}$ | $73.98_{\pm.02}$ | $73.83_{\pm.04}$ | $73.07_{\pm.01}$ | $75.08_{\pm.82}$ | $75.33_{\pm.02}$ |
| | F | $68.04_{\pm.03}$ | $64.05_{\pm.04}$ | $64.84_{\pm.04}$ | $64.06_{\pm.04}$ | $65.51_{\pm.02}$ | $64.02_{\pm.04}$ | $63.11_{\pm.02}$ | $62.59_{\pm.04}$ | $63.77_{\pm.06}$ | $63.58_{\pm.06}$ | $64.36_{\pm.02}$ | $66.13_{\pm1.38}$ | $66.89_{\pm.02}$ |
| **3 RND** | E | $59.56_{\pm.06}$ | $57.53_{\pm.06}$ | $57.41_{\pm.06}$ | $56.38_{\pm.11}$ | $57.76_{\pm.05}$ | $58.83_{\pm.10}$ | $54.41_{\pm.13}$ | $58.07_{\pm.12}$ | $58.14_{\pm.04}$ | $57.46_{\pm.10}$ | $57.55_{\pm.03}$ | $58.85_{\pm.57}$ | $59.09_{\pm.05}$ |
| | M | $68.22_{\pm.04}$ | $65.86_{\pm.03}$ | $66.29_{\pm.03}$ | $65.34_{\pm.06}$ | $67.03_{\pm.03}$ | $68.62_{\pm.05}$ | $65.54_{\pm.06}$ | $64.98_{\pm.08}$ | $67.34_{\pm.04}$ | $67.11_{\pm.06}$ | $66.69_{\pm.02}$ | $68.18_{\pm.38}$ | $68.24_{\pm.03}$ |
| | H | $75.75_{\pm.02}$ | $72.66_{\pm.02}$ | $72.42_{\pm.03}$ | $72.00_{\pm.03}$ | $73.52_{\pm.02}$ | $75.63_{\pm.03}$ | $73.36_{\pm.03}$ | $69.30_{\pm.06}$ | $74.04_{\pm.02}$ | $75.36_{\pm.03}$ | $73.40_{\pm.01}$ | $75.58_{\pm.17}$ | $75.39_{\pm.01}$ |
| | F | $67.72_{\pm.04}$ | $64.98_{\pm.02}$ | $65.31_{\pm.04}$ | $64.45_{\pm.04}$ | $66.17_{\pm.02}$ | $67.54_{\pm.04}$ | $64.36_{\pm.06}$ | $64.33_{\pm.03}$ | $66.42_{\pm.03}$ | $66.23_{\pm.04}$ | $65.75_{\pm.02}$ | $67.23_{\pm.58}$ | $67.34_{\pm.03}$ |
| **4 PGD** | E | $59.70_{\pm.06}$ | $57.71_{\pm.05}$ | $57.73_{\pm.09}$ | $57.19_{\pm.07}$ | $57.60_{\pm.08}$ | $57.05_{\pm.17}$ | $54.69_{\pm.09}$ | $58.18_{\pm.07}$ | $58.27_{\pm.09}$ | $58.46_{\pm.11}$ | $57.66_{\pm.05}$ | $58.81_{\pm.64}$ | $59.14_{\pm.05}$ |
| | M | $68.40_{\pm.05}$ | $66.12_{\pm.02}$ | $66.39_{\pm.04}$ | $65.67_{\pm.04}$ | $67.04_{\pm.03}$ | $68.24_{\pm.04}$ | $65.64_{\pm.08}$ | $65.17_{\pm.06}$ | $67.32_{\pm.03}$ | $67.85_{\pm.05}$ | $66.78_{\pm.02}$ | $68.16_{\pm.23}$ | $68.12_{\pm.03}$ |
| | H | $75.83_{\pm.04}$ | $72.91_{\pm.02}$ | $72.47_{\pm.04}$ | $72.18_{\pm.05}$ | $73.52_{\pm.02}$ | $75.55_{\pm.05}$ | $73.58_{\pm.04}$ | $69.64_{\pm.03}$ | $73.89_{\pm.02}$ | $74.34_{\pm.04}$ | $73.39_{\pm.01}$ | $75.24_{\pm.65}$ | $75.36_{\pm.02}$ |
| | F | $68.01_{\pm.02}$ | $65.41_{\pm.01}$ | $65.54_{\pm.03}$ | $65.05_{\pm.04}$ | $66.22_{\pm.02}$ | $66.49_{\pm.04}$ | $64.63_{\pm.04}$ | $64.82_{\pm.04}$ | $66.32_{\pm.02}$ | $66.14_{\pm.04}$ | $65.86_{\pm.01}$ | $66.94_{\pm.76}$ | $67.37_{\pm.02}$ |
| **5 FGSM** | E | $59.71_{\pm.05}$ | $57.69_{\pm.08}$ | $57.62_{\pm.06}$ | $57.16_{\pm.08}$ | $57.60_{\pm.06}$ | $56.97_{\pm.09}$ | $54.67_{\pm.08}$ | $58.20_{\pm.10}$ | $58.23_{\pm.06}$ | $58.46_{\pm.07}$ | $57.63_{\pm.05}$ | $58.81_{\pm.65}$ | $59.15_{\pm.04}$ |
| | M | $68.37_{\pm.02}$ | $66.10_{\pm.03}$ | $66.38_{\pm.04}$ | $65.70_{\pm.05}$ | $67.03_{\pm.04}$ | $68.27_{\pm.04}$ | $65.61_{\pm.08}$ | $65.16_{\pm.05}$ | $67.30_{\pm.02}$ | $67.84_{\pm.07}$ | $66.77_{\pm.02}$ | $68.16_{\pm.23}$ | $68.11_{\pm.02}$ |
| | H | $75.82_{\pm.02}$ | $72.92_{\pm.04}$ | $72.48_{\pm.03}$ | $72.18_{\pm.04}$ | $73.52_{\pm.02}$ | $75.55_{\pm.05}$ | $73.60_{\pm.04}$ | $69.64_{\pm.04}$ | $73.90_{\pm.01}$ | $74.34_{\pm.04}$ | $73.39_{\pm.01}$ | $75.23_{\pm.65}$ | $75.35_{\pm.02}$ |
| | F | $68.00_{\pm.02}$ | $65.41_{\pm.02}$ | $65.54_{\pm.04}$ | $65.05_{\pm.04}$ | $66.22_{\pm.02}$ | $66.50_{\pm.06}$ | $64.65_{\pm.04}$ | $64.82_{\pm.03}$ | $66.34_{\pm.03}$ | $66.15_{\pm.06}$ | $65.95_{\pm.05}$ | $66.94_{\pm.76}$ | $67.37_{\pm.01}$ |
| **6 W/O Attack** | E | $59.67_{\pm.00}$ | $58.08_{\pm.00}$ | $60.22_{\pm.00}$ | $58.53_{\pm.00}$ | $58.14_{\pm.00}$ | $60.78_{\pm.00}$ | $56.83_{\pm.00}$ | $59.47_{\pm.00}$ | $59.62_{\pm.00}$ | $59.88_{\pm.00}$ | $59.12_{\pm.00}$ | $60.29_{\pm.37}$ | $60.42_{\pm.00}$ |
| | M | $68.28_{\pm.00}$ | $66.14_{\pm.00}$ | $67.11_{\pm.00}$ | $66.35_{\pm.00}$ | $67.00_{\pm.00}$ | $68.98_{\pm.00}$ | $66.26_{\pm.00}$ | $65.45_{\pm.00}$ | $67.53_{\pm.00}$ | $68.41_{\pm.00}$ | $67.15_{\pm.00}$ | $68.56_{\pm.30}$ | $68.59_{\pm.00}$ |
| | H | $75.85_{\pm.00}$ | $73.05_{\pm.00}$ | $72.69_{\pm.00}$ | $72.66_{\pm.00}$ | $73.46_{\pm.00}$ | $75.64_{\pm.00}$ | $73.69_{\pm.00}$ | $69.84_{\pm.00}$ | $74.10_{\pm.00}$ | $75.76_{\pm.00}$ | $73.67_{\pm.00}$ | $75.75_{\pm.09}$ | $75.52_{\pm.00}$ |
| | F | $67.93_{\pm.00}$ | $65.76_{\pm.00}$ | $66.68_{\pm.00}$ | $65.85_{\pm.00}$ | $66.20_{\pm.00}$ | $68.47_{\pm.00}$ | $65.59_{\pm.00}$ | $64.91_{\pm.00}$ | $67.08_{\pm.00}$ | $68.02_{\pm.00}$ | $66.65_{\pm.00}$ | $68.14_{\pm.24}$ | $68.11_{\pm.00}$ |
| **Avg. Accuracy** | E | $59.62_{\pm.02}$ | $57.44_{\pm.03}$ | $57.77_{\pm.03}$ | $56.84_{\pm.04}$ | $56.79_{\pm.04}$ | $55.62_{\pm.06}$ | $54.33_{\pm.04}$ | $57.53_{\pm.05}$ | $56.74_{\pm.00}$ | $55.67_{\pm.10}$ | - | - | - |
| | M | $68.34_{\pm.01}$ | $65.88_{\pm.01}$ | $66.41_{\pm.01}$ | $65.59_{\pm.02}$ | $66.94_{\pm.02}$ | $67.24_{\pm.19}$ | $65.43_{\pm.03}$ | $64.97_{\pm.02}$ | $67.28_{\pm.01}$ | $66.85_{\pm.18}$ | - | - | - |
| | H | $75.84_{\pm.01}$ | $72.69_{\pm.01}$ | $72.42_{\pm.01}$ | $72.14_{\pm.02}$ | $73.47_{\pm.01}$ | $75.49_{\pm.01}$ | $73.33_{\pm.02}$ | $69.38_{\pm.02}$ | $73.98_{\pm.00}$ | $74.78_{\pm.02}$ | - | - | - |
| | F | $67.90_{\pm.01}$ | $64.87_{\pm.03}$ | $65.02_{\pm.03}$ | $64.41_{\pm.04}$ | $65.12_{\pm.25}$ | $65.45_{\pm.26}$ | $63.65_{\pm.07}$ | $63.42_{\pm.29}$ | $64.53_{\pm.84}$ | $64.46_{\pm1.13}$ | - | - | - |
| **Avg. 3-Min Accuracy** | E | $59.55_{\pm.03}$ | $57.05_{\pm.04}$ | $57.02_{\pm.03}$ | $56.05_{\pm.07}$ | $55.73_{\pm.07}$ | $52.33_{\pm.12}$ | $53.25_{\pm.07}$ | $56.43_{\pm.07}$ | $54.77_{\pm.16}$ | $54.32_{\pm.17}$ | - | - | - |
| | M | $68.28_{\pm.01}$ | $65.64_{\pm.02}$ | $66.20_{\pm.01}$ | $65.28_{\pm.03}$ | $66.84_{\pm.02}$ | $65.85_{\pm.40}$ | $65.02_{\pm.04}$ | $64.69_{\pm.03}$ | $67.17_{\pm.02}$ | $65.66_{\pm.34}$ | - | - | - |
| | H | $75.80_{\pm.02}$ | $72.42_{\pm.02}$ | $72.29_{\pm.01}$ | $71.93_{\pm.02}$ | $73.42_{\pm.01}$ | $75.36_{\pm.02}$ | $73.05_{\pm.02}$ | $69.04_{\pm.03}$ | $73.92_{\pm.01}$ | $74.17_{\pm.03}$ | - | - | - |
| | F | $67.78_{\pm.02}$ | $64.22_{\pm.11}$ | $64.12_{\pm.06}$ | $63.50_{\pm.08}$ | $64.02_{\pm.49}$ | $63.39_{\pm.53}$ | $62.35_{\pm.14}$ | $61.99_{\pm.58}$ | $62.44_{\pm1.69}$ | $62.11_{\pm2.26}$ | - | - | - |
| **Weighted Accuracy** | E | $59.53_{\pm.04}$ | $56.93_{\pm.04}$ | $56.94_{\pm.04}$ | $55.93_{\pm.08}$ | $54.63_{\pm.14}$ | $48.21_{\pm.27}$ | $52.23_{\pm.08}$ | $55.55_{\pm.14}$ | $52.18_{\pm.33}$ | $47.45_{\pm.35}$ | - | - | - |
| | M | $68.25_{\pm.02}$ | $65.57_{\pm.02}$ | $66.17_{\pm.02}$ | $65.28_{\pm.02}$ | $66.79_{\pm.02}$ | $63.85_{\pm.80}$ | $64.77_{\pm.07}$ | $64.60_{\pm.03}$ | $67.06_{\pm.03}$ | $64.07_{\pm.68}$ | - | - | - |
| | H | $75.78_{\pm.02}$ | $72.37_{\pm.02}$ | $72.20_{\pm.01}$ | $71.92_{\pm.03}$ | $73.41_{\pm.01}$ | $75.30_{\pm.02}$ | $72.98_{\pm.02}$ | $68.99_{\pm.04}$ | $73.91_{\pm.01}$ | $74.08_{\pm.03}$ | - | - | - |
| | F | $67.73_{\pm.03}$ | $63.96_{\pm.21}$ | $63.19_{\pm.10}$ | $62.80_{\pm.15}$ | $62.18_{\pm.98}$ | $61.58_{\pm1.05}$ | $61.00_{\pm.28}$ | $60.54_{\pm1.18}$ | $59.82_{\pm3.38}$ | $59.37_{\pm4.53}$ | - | - | - |