# OpenReview forum: "Graph Robustness Benchmark: Benchmarking the Adversarial Robustness of Graph Machine Learning"
_NeurIPS.cc/2021/Track/Datasets_and_Benchmarks/Round2 — NeurIPS 2021 Datasets and Benchmarks Track (Round 2)_

### Official Review · Reviewer_59mL · 2021-09-21
**A unified benchmark for graph neural networks.**

**Rating:** 7
**Confidence:** 5
**Correctness:** Yes, the claims made in the submissio…
**Clarity:** Yes, the paper is well written.

**Strengths:**

This paper provides a very rigorous benchmarking of the robustness of graph neural networks. The benchmark is well documented and leaderboards are easily accessible (both in paper and on the webpage). The paper is also well written and easily accessible.

**Weaknesses:**

While the proposed framework is extendible to any dataset, I encourage authors to expand the current analysis to multiple datasets in the paper itself. Currently all major results are reported mostly on grb-aminer dataset (from figure 4,5,6,7,10,11, table 1) which natural raises the question: whether the findings are consistent across datasets.

Similarly the current benchmark remain focused only on node-classification. I encourage authors to at least provide a preliminary sketch or detailed discussion on how they are going to handle the expansion of current benchmark. Being forward looking is essential for a benchmark to be well-established.

Additionally, I still find the evaluation on normalization and adversarial training to improve the robustness pretty thin. While authors claim their results being state-of-the-art, there are multiple questions left unanswered: 1) is this phenomenon specific to layer normalization or its extendible to other normalization techniques too 2) What will be the performance impact with different variations of adversarial training, etc. I encourage authors to rethink this aspect.




**Additional Feedback:**

I've already discussed it in an earlier section.

**Documentation:**

Yes, documentation is available.

**Relation To Prior Work:**

Yes, authors provide adequate references to the related work.

**Summary And Contributions:**

This paper does a unified benchmarking of graph neural networks with a key focus on different attacks vectors, datasets and models. Additionally, the proposed benchmark is modular, thus it supports multiple different types of attack vectors, external datasets, graph development frameworks.

---

> ### Author Response · Authors · 2021-09-29
> **About more experimental analysis and expansion of the benchmark.**
>
> Thanks very much for your valuable advice. Here are our responses that address your concerns:
>
> 1. **Expansion of the current analysis to multiple datasets:** We have updated Figure
> 4, 5, 6, 7, 10, 11, and Table 13-17 for all five datasets in the revised version. We think the question "whether the findings are consistent across datasets" is a very valuable one. The results show that the robustness of the methods is actually related to the graph data. The rankings of robustness may vary a lot across datasets. There are similar situations in other graph benchmarks. For example, in Open Graph Benchmark (OGB), there doesn't exist a dominant architecture for all datasets. Certain model architecture only works for a certain dataset. We think that it is because graphs are actually quite divergent. Thus, we suggest **when giving conclusions about robustness in GML, one should not only consider the model itself but also take the graph data into account.** We discuss this phenomenon in Section 4 (Line 302-318). It still remains a mystery "whether there exist GML models that are robust for any kind of graph?" GRB provides tools to investigate it and heads a very small step forwards.
>
> 2. **Expansion to other tasks:** We added a Maintenance Plan in Section 7 that discusses the future plan of GRB. Benefit from the modular design, GRB can be easily extended to other tasks. To give a concrete example of how to do this, we have added supports for graph classification tasks, here are some [examples](https://github.com/THUDM/grb/tree/master/examples/graph_classification) (tutorials based on notebook showing details of how GRB can be used for graph classification). For new tasks, it requires adding task-specific functions in each module (dataset, model, trainer, attack, defense, etc.). Meanwhile, some functions are common in GML thus can be reused for different tasks.
> Apart from implementations, benchmarking new tasks need to carefully define the threat model and the evaluation process, which we are still working on. In the future, as more researches cover the robustness of other GML tasks, the consensus might be easier reached by the community and our benchmark will take it into account.
>
> 3. **About layer normalization and adversarial training:** Considering the specificity of this track, we focus more on designing the benchmark rather than analyze specific techniques. The initial motivation of adding these two general methods is (1) Previous defenses like GNN-SVD or GNNGUARD are not scalable for large-scale graphs, thus we need better baselines. (2) The idea behind LN and AT is simple and can be generally adapted to GML models. We agree that there might be better variations, however, this is not the emphasis of this work. We appreciate your advice and will leave unanswered questions as future works. More details of these methods can be found in the Appendix (Line 704-732), and the documented implementations can be found in our Github repo.

---

> > ### Comment · Reviewer_59mL · 2021-09-29
> > **Significant update in experimental section**
> >
> > Thanks for expanding the experimental section to include more datasets. It indeed makes it more rigorous and naturally leads to a new observation, i.e., no single model architecture is dominant on all datasets. I've updated my score accordingly.

---

### Official Review · Reviewer_GHNt · 2021-09-21
**A potentially misleading benchmark**

**Rating:** 4
**Confidence:** 4
**Clarity:** Yes.

**Strengths:**

The existence of agreed-upon datasets and threat models for evaluating GML models certainly seems useful. Having a _library_ of attacks and defenses seems useful as well.

Turning this into a _benchmark_ is the tricky part (more details below).


**Weaknesses:**

The existence of benchmarks in ML is certainly useful, but when benchmarking models in a security context, this fundamentally cannot be done in a fully automated way. An analog would be to compare to software security: we can't measure if software is secure by running all past CVEs on the software and checking if the attacks succeed. Running existing attacks on defenses only gives an upper bound on robustness. Usually, to get tighter bounds on robustness, adaptive attacks are needed (the right thing to do in a security context). The existence of GRB in its current form is dangerous and could mislead researchers: the benchmark does not measure robustness, it simply gives _upper bounds_ for robustness, and even there, the bounds are not necessarily comparable across defenses (the existing attacks may happen to work better on defense A compared to defense B, but it may be the case that against better tuned attacks, defense A outperforms defense B), so it's hard to draw any deep conclusions from benchmarking results. That being said, I don't think it's not possible to have a reasonable benchmark for robustness. One way to do it would be to have a leaderboard where third-party researchers can submit results from adaptive attacks (e.g. like robust-ml.org). Another semi-automated approach would be to follow something like RobustBench (robustbench.github.io): have a suite of attacks that give a rough upper bound for robustness (and even there, having a thorough argument for why the attacks give reasonable bounds for the admitted defense models), and _also_ allow third-party researchers to submit adaptive attacks on models in the leaderboard.

Some more detailed points:

- The paper's definition of black-box attackers is unreasonable. See prior work like https://arxiv.org/abs/1902.06705 and Kerckhoffs's principle. It is not possible/reasonable to model an adversary that has no knowledge of the defense mechanism. (Though it may be reasonable to keep model _parameters_ secret.)
- The leaderboard reports accuracy for each model on each attack. This is fine data to report, but when considering robustness overall, a single number better represents robustness (the attacker runs all the attacks and uses the best result for each data point; under this attack, what is the model accuracy). Again, see prior work arguing that this is the right metric to use. Weighted accuracy of multiple attacks seems like a strange metric.
- For attacks that have parameters (e.g. step size), how are these parameters chosen? Are they tuned per model being attacked? This is another factor that could lead to a misleading leaderboard: when parameter choice makes one defense work better than another.


**Additional Feedback:**

A robustness benchmark for graph machine learning certainly seems useful to have! GRB is heading in the right direction, but I think it is critical to address the weaknesses above before such a benchmark is relied upon by the research community.


**Correctness:**

I don't think there's anything _incorrect_ said in the paper per se, but the benchmark design has some serious flaws (more details in the weaknesses section).


**Documentation:**

Yes.

**Ethics:**

N/A. The paper discusses in Section 6 a potential negative impact of making attack methods more widely known, but this is not a real concern; the security community has accepted that security research can aid malicious actors, but the benefits of performing and publishing security research outweigh the costs.


**Relation To Prior Work:**

The paper would benefit from comparison to prior work on benchmarking robustness in other domains.


**Summary And Contributions:**

This paper contributes the Graph Robustness Benchmark (GRB), measuring the robustness to adversarial examples of graph machine learning models. The benchmark standardizes threat models, datasets, defenses, and attacks, making it easy to run many attacks against many defenses in a unified evaluation scenario.

---

> ### Author Response · Authors · 2021-09-29
> **Response 1/2: A potentially misleading review.**
>
> Thanks the reviewer for sharing thoughts derived from domains like software security and image classification. However, we would like to emphasize that the focus of this work is on the adversarial attack and defense in Graph Machine Learning (GML), which has different problem settings and developments. The setting in GRB follows the common setup in the field of GML attacks and defenses, such as those in ACM KDD CUP 2020 [1], KDD 2018 Best Paper [2], and recent machine learning publications [3,4,5].
>
> We will adopt the reviewer's suggestion to discuss and potentially learn from the difference between GRB and works from other domains, but we would like to clarify some potential misunderstandings:
>
> 1. **Reviewer defines our work as measuring robustness, which is not exactly what we do.**
>
> Given the different problem setups and domains, it may be inappropriate to make a direct comparison between RobustBench (for image classification) and GRB.
> The goal of RobustBench is to measure a rough bound by giving a single number that represents robustness, to achieve which it considers a white-box scenario and tries as hard as possible to attack every image sample.
> However, we'd like to answer a more general and practical question in GML, i.e., "How can we choose GML models that are generally more robust against potential attacks on a specific type of graph data?”.
> For this purpose, GRB has threat models that are dedicated to GML, which differs from the one considered in RobustBench.
>
>
> 2. **Black-box vs. white-box.**
>
> The reviewer considers the black-box scenario unreasonable, while the white-box scenario can give a rough upper bound of robustness as RobustBench does.
> However, as explained previously, the focus of GRB is not on determining the promising bound for robustness.
> For GML applications (e.g., recommender systems, social networks), black-box is in practice the most common scenario, where the information of models is hidden secretly. it would raise a big threat if a black-box attack could still work under limited knowledge.
> A notable example can be found in ACM KDD CUP 2020, which hosted a black-box challenge about “Adversarial attacks and defenses on academic graphs” [1].
> In addition, previous GML attack studies, including the KDD 2018 best paper [2] and other ML publications [3] [4], also consider the black-box (evasion) attack setting.
>
> Furthermore, the reviewer suggests to check Kerckhoffs’ principle. Note that this is actually discussed in Section 3 (Line 191).
> Without knowing the underlying model deployed (for example, those graph learning models in industry, e.g., Facebook, Twitter, and Alibaba), attackers can make assumptions like using the most popular GML models as the surrogate model, or adding potential defense mechanisms.
> However, a totally white-box scenario, in which the attacker knows all details of the model and "runs all the attacks and uses the best result for each data point", might benefit the search of the theoretical bound of robustness, but might not be of practical interest and use to GML.
>
> Thus, to best advocate the practice usage of GRB and follow the GML community convention, we believe that the black-box scenario is reasonable at the current stage.
> Specifically, we accept new submissions (including adaptive attacks) through [google form](https://docs.google.com/forms/d/e/1FAIpQLSfJaUK-SXYFnlSqTEEwTOwsqzA5JnpXyvZe8E24hlLE7scRcA/viewform).
> We have [Rules](https://cogdl.ai/grb/intro/rules) and [Examples](https://github.com/THUDM/grb/tree/master/examples) for new submissions to follow to ensure they don’t use unreasonable prior knowledge.
> We will also host a leaderboard that evaluates submissions under a strict black-box scenario. We welcome researchers to show the possibility of how to obtain hidden information in this case.

---

> ### Author Response · Authors · 2021-09-29
> **Response 2/2: A potentially misleading review.**
>
> 3. **Choice of weighted accuracy.**
>
> This setting has been well-motivated and adopted by KDD CUP 2020 and recent studies.
> Here is the reasoning:
> In order to fairly compare multiple defenses and attacks, a unified evaluation is required.
> The weighted accuracy satisfies the requirement by attaching higher weights to more effective attacks (A higher value means that the model is still robust under all potential attacks).
> Note that by design this is not a fixed number and is to be updated when there are more powerful attacks (since the ranking of attacks will change, thus the weight attached to the specific method is dynamically updated).
> In GML, there is thus far no dominant model across different graph datasets (this can be observed from the [Stanford OGB leaderboard](https://ogb.stanford.edu/), on which the performance of a single GML model architecture varies a lot on different graphs.
> Thus, it is less meaningful to give a fixed robustness measure for a single model.
> The reviewer mentioned “the existing attacks may happen to work better on defense A compared to defense B, but it may be the case that against better-tuned attacks, defense A outperforms defense B”, the adopted metric handles this situation since we consider a more general case of multiple methods.
> If one designs an attack to specifically target defense A, and it is likely that it might be less effective to defense B. The ranking of defenses changes, the weighted accuracy changes as well.
> Therefore, the weighted accuracy metric can be used to tell whether an attack is generally effective (Note that in GML applications, the usage of an ensemble of various models is very common).
>
>
> 4. **Choice of the attack parameters.**
>
> Since the scenario is black-box, the attack can not be tuned per model (which only happens in white-box). The current attack process is: For gradient-based attacks, the attacker first trains a surrogate model, then tunes attack parameters on it to generate the attacked graph. For gradient-free attacks, the attacked graph is generated based on graph properties. Finally the attacked graph is transferred to defense models. Note that the attacker can choose any surrogate model or even add defense mechanism to it, but without knowing the exact parameters of the target model. However, since the weighted metric evaluates the overall performance, tuned on a specific or randomly-chosen model might not be effective. For now, we follow the finding in [5], which suggests that choosing the vanilla GCN as the surrogate model has a better transferability. This design choice can and should be updated in GRB if new models are demonstrated as better surrogate model options.
>
> In summary, we would like to argue that GRB is by design for benchmarking the attack and defense strategies in graph machine learning. Though we should learn from benchmarks from other domains, GRB has fundamentally different setups and goals from them.
>
>
> References
>
> [1] ACM KDD CUP 2020 "Graph Adversarial Attack and Defense on Academic Graphs." https://www.biendata.xyz/competition/kddcup_2020_formal/
>
> [2] Daniel Zügner, Amir Akbarnejad, and Stephan Günnemann. "Adversarial attacks on neural networks for graph data." **KDD 2018 Best Paper Award.**
>
> [3] Hanjun Dai, et al. "Adversarial attack on graph structured data." ICML 2018.
>
> [4] Lichao Sun, et al. "Adversarial attack and defense on graph data: A survey." arXiv preprint arXiv:1812.10528 (2018).
>
> [5] Xu Zou, et al. “TDGIA: Effective injection attacks on graph neural networks.” KDD 2020.
>
> [6] Weihua Hu, et al. "Open graph benchmark: Datasets for machine learning on graphs.” NeurIPS 2020.

---

> > ### Comment · Reviewer_GHNt · 2021-10-04
> > **Response**
> >
> > 1. Measuring robustness
> >
> > I am unconvinced of the importance of knowing which models are "generally more robust" against non-adaptive attacks in the security context, especially because it's possible/easy to "break" existing attacks, even accidentally, giving the appearance of robustness without having true robustness. I would understand if the intention was to measure robustness in a non-security context, like measuring robustness to "random corruptions" (e.g. like work on image corruptions), but once we're talking about adversarial attacks in a security context where there is an attacker and the attacker can differentiate through the model, "general robustness" seems like an odd in-between notion.
> >
> > 2. Black-box vs white-box
> >
> > On the topic of black-box attacks: making precise what assumptions are allowed is the tricky part. The rules say that it's reasonable to assume the use of GCNs. Is the adversary allowed to guess the model architecture? How "close" is the guess allowed to be to the model under attack? To make a comparison to CNNs since they have better-known examples: would it be reasonable to attack a ResNet-150 using a ResNet-50 surrogate model? What about an NasNet model? What about a ResNet-150 model? Where do you draw the line in modeling "lack of knowledge" / "reasonable assumptions"?
> >
> > 3. Choice of weighted accuracy
> >
> > I'm arguing that it doesn't make sense to compare multiple defenses and attacks. It seems reasonable to compare defenses/models: what is the robustness under a given threat model of a GCN model? Whereas for attacks, it's strange to ask the question of "how well does this attack work on average across models?", especially with no per-model tuning/customization.
> >
> > 4. Choice of attack parameters
> >
> > > Since the scenario is black-box, the attack can not be tuned per model
> >
> > This depends on the definition of black-box. If we consider a real-world scenario where the adversary can at least query the model under attack, it's possible for the adversary to tune the attack to the model. If we define black-box as only including transfer-based attacks, and ones that cannot check the results and potentially try a different surrogate model, or other strategies, then yes, the attack definitionally cannot be tuned per model. I think such a threat model is not that interesting / not that realistic.

---

### Official Review · Reviewer_zT1n · 2021-09-21
**A Clear and Promising Benchmark for Adversarial Attacks in GML**

**Rating:** 7
**Confidence:** 3

**Strengths:**

The work presented by the authors has a clear use in the field of GraphML, and the engineering work the authors have put in to it lowers the amount of effort other researchers need to spend designing similar experiments, freeing up more time to focus on the actually development and advancement of academic work. The authors do a good job of making sure their work is accessible, their website is clear and informative, the data is easily presented and accessible, and the documentation is sufficient to allow a researcher fairly familiar with the field to make use of it without excessive overhead.

**Weaknesses:**

The work does have some limitations that limit it's relevancy and extensibility, for example:
1. The authors only consider the task of node classification. The authors themselves note this limitation and I am hopeful that the authors will extend this work in the future, but at this work can only be used for one task.
2. It is unclear exactly how the authors plan to maintain and extend this work. The authors mention that they'll be updating the leaderboard, which is great, but there are other facets to consider. Given that this is a benchmark for focusing on adversarial attack and defense, this benchmark must surely be updated every so often with the newest relevant attacks to make sure that the evaluations provided are relevant There are many ways the authors could propose to maintain their work reasonably current, but at the time of writing this review, I do not know exactly how the authors plan to keep current which may limit the usefulness of this work in the future.

**Additional Feedback:**

Overall I am positive about this work, my main concerns are as follows:
1. Currently, the work is limited in scope (only a single task and a few attacks that are being evaluated). How do the authors
   1. Plan to expand the scope of their work?
   2. Plan to keep their benchmark current and useful with the state of the art attacks?
2. I am a bit concern that due to the authors choice of splits may hinder generalization to real world scenarios and as such may impact how useful this work actually is. Can the authors either provide an argument for why my concern is unlikely to ever materialize or matter, or could they add in some more splits to test their data on to ensure that anything observed in this work generalizes?

**Clarity:**

The figures and the majority of the paper are clear enough, however I encountered some grammatical issues while reading through this work. For example:
Line 80, this is not the proper way to cite something, it should've been something like "In the work of Xu et al [32]" or similar

Line 115, "Facing the challenges" -> "Facing these challenges"

Not an exhaustive list and while these didn't make the work ambiguous, I would still like to see the authors take another pass at editing their work.

**Correctness:**

For the most part, I don't have many concerns with the approach the authors detailed. However, I am a bit concerned by the splitting methodology in use by the authors.

I take the authors point that a single random split may not be suitable (as they cite from Shcur et al). However as I recall, the Shcur et al propose that a single spit on it's own can be problematic, and to my understanding while the authors define their own splitting scheme, they only have a single split which sort of goes against the work they are citing. To avoid problems where certain methods work better or worse just due to the spits used and nothing else, shouldn't the authors generate a set of splits and aggregate results over those splits instead of a single split?

My second correctness concern still has to do with the splitting. I'm worried about a case where the fact that the authors use this particular splitting schema (which to my knowledge hasn't been commonly used) results in results that may not generalize to how GML models are used in practice. I understand that the authors choose these schema to try to better evaluate the adversarial methods, but the largest use for a benchmark of this sort is to generate information that can be used to generalize about which attacks are worth worrying about or how to defend from those attacks. By using these splits, I can imagine a scenario where now the training and test distributions are sufficiently different from what other methods use that the information generated from this benchmark ceases to be useful. Do the authors anticipate this to be the case?

**Documentation:**

The authors lack a concrete maintenance and update plan, the on all other the fronts the authors do very well.

**Relation To Prior Work:**

The authors do a really good job of this, they have high level overviews of how this work incorporates previous work in the mai text and a much more thorough explanation for the attacks and defenses that the authors implement is in the supplemental material.

**Summary And Contributions:**

The Authors Present Graph Robustness Benchmark (GRB), a benchmark that aims to provide a standardized evaluation framework for measuring attacks and defenses in adversarial GML on the node classification task. The authors present a benchmark that is:
* Clear and well thought
* Aims to provide a standard and extensible way to examine the performance of various attack and defense methods
* Provides ways to see how the attack and defense methods generalize to different scales and datasets
* Present a unified framework to evaluate these methods, including a clear website leaderboard and documentation

---

> ### Author Response · Authors · 2021-09-29
> **Response 1/2: About extension and maintenance of the benchmark.**
>
> Thanks very much for your insightful comments. Here are our responses that address your concerns:
>
> 1. **Extension of tasks:** The current work focuses on node classification, the main reason is that the majority of previous works about robustness in GML study this typical task. Even though other tasks (e.g. graph classification, link prediction) are rarely studied in the scope of robustness, we agree that it is also essential to guarantee robustness in those tasks. This requires high extensibility of the benchmark, that's exactly why we build GRB in a modular design rather than dirty codes. Benefit from this design, it can be easily extended to other tasks. We have added some [examples](https://github.com/THUDM/grb/tree/master/examples/graph_classification) based on notebook showing details of how GRB can be used for graph classification. For new tasks, it requires adding task-specific functions in each module (dataset, model, trainer, attack, defense, etc.). Meanwhile, common functions in GML can be reused for different tasks.
> Apart from implementations, benchmarking new tasks need thorough consideration of the threat model and the evaluation process, which we are still working on. In the future, as more researches cover the robustness of other GML tasks, the consensus might be easier reached by the community and our benchmark will take it into account.
>
> 2. **Maintenance plan:** We have added the maintenance plan (Section 7) in the revised version. The goals of GRB are (1) providing a unified benchmark that various methods can be fairly compared; (2) facilitating the implementation thus researchers can focus on investigating truly valuable solutions; (3) tracking the progress of robustness in GML.
>
>       To achieve these goals, we have a long-term maintenance plan: (1) We will regularly include SOTA methods to GRB by updating the "method zoo". To make the community easier to contribute GRB, we allow submissions through [google form](https://docs.google.com/forms/d/e/1FAIpQLSfJaUK-SXYFnlSqTEEwTOwsqzA5JnpXyvZe8E24hlLE7scRcA/viewform). Researchers can provide new attacks or defenses, following the [examples](https://github.com/THUDM/grb/tree/master/examples) and [rules](https://cogdl.ai/grb/intro/rules) that we provided in detail. (2) Based on these new submissions, we will verify their correctness and update the results on leaderboards. They give a clear view of the performance of new methods compared with the old ones. (3) As the domain gradually develops, there might be a better understanding of defining the threat model or doing the evaluation. It is important for us to continuously revise the settings of GRB and make proper improvements (like including new scenarios or more constraints). All of the above measures will ensure the usefulness of GRB.

---

> ### Author Response · Authors · 2021-09-29
> **Response 2/2: About the proposed splitting scheme.**
>
> 3. **Design of the splitting methodology:** A single totally random split is problematic, which might lead to less general results as Shchur et al propose. However, in our design, the splitting still follows a specific degree distribution of the graph rather than being entirely random, as shown in Figure 4. To show the generalization of the splitting scheme, we did tests on grb-cora datasets. We create 10 splittings using the designed scheme, and train GML models 10 times for each split.
>
> | Seed | GCN                 | SAGE                | SGCN                | TAGCN               | APPNP               | GIN                 | GAT                 |
> | :--: | :------------------ | :------------------ | :------------------ | :------------------ | :------------------ | :------------------ | :------------------ |
> | 0    | 0\.8424$\pm$0.0041 | 0\.7800$\pm$0.0142 | 0\.8459$\pm$0.0055 | 0\.8408$\pm$0.0078 | 0\.7902$\pm$0.0404 | 0\.8183$\pm$0.0114 | 0\.8453$\pm$0.0082 |
> | 1    | 0\.8775$\pm$0.0047 | 0\.8195$\pm$0.0135 | 0\.8761$\pm$0.0046 | 0\.8606$\pm$0.0086 | 0\.8230$\pm$0.0315 | 0\.8388$\pm$0.0129 | 0\.8677$\pm$0.0102 |
> | 2    | 0\.8496$\pm$0.0079 | 0\.7864$\pm$0.0124 | 0\.8586$\pm$0.0069 | 0\.8277$\pm$0.0099 | 0\.8193$\pm$0.0270 | 0\.8000$\pm$0.0097 | 0\.8500$\pm$0.0040 |
> | 3    | 0\.8577$\pm$0.0065 | 0\.8091$\pm$0.0162 | 0\.8648$\pm$0.0070 | 0\.8410$\pm$0.0106 | 0\.8325$\pm$0.0408 | 0\.8418$\pm$0.0091 | 0\.8611$\pm$0.0077 |
> | 4    | 0\.8622$\pm$0.0043 | 0\.8032$\pm$0.0116 | 0\.8611$\pm$0.0057 | 0\.8495$\pm$0.0057 | 0\.8366$\pm$0.0493 | 0\.8204$\pm$0.0071 | 0\.8619$\pm$0.0106 |
> | 5    | 0\.8631$\pm$0.0044 | 0\.8080$\pm$0.0148 | 0\.8649$\pm$0.0100 | 0\.8485$\pm$0.0088 | 0\.8437$\pm$0.0331 | 0\.8410$\pm$0.0084 | 0\.8648$\pm$0.0086 |
> | 6    | 0\.8480$\pm$0.0089 | 0\.8149$\pm$0.0075 | 0\.8619$\pm$0.0027 | 0\.8352$\pm$0.0070 | 0\.7973$\pm$0.0618 | 0\.8260$\pm$0.0046 | 0\.8537$\pm$0.0082 |
> | 7    | 0\.8736$\pm$0.0058 | 0\.8005$\pm$0.0177 | 0\.8674$\pm$0.0031 | 0\.8621$\pm$0.0047 | 0\.8649$\pm$0.0211 | 0\.8356$\pm$0.0121 | 0\.8684$\pm$0.0086 |
> | 8    | 0\.8430$\pm$0.0062 | 0\.8096$\pm$0.0078 | 0\.8486$\pm$0.0087 | 0\.8469$\pm$0.0083 | 0\.8289$\pm$0.0244 | 0\.8317$\pm$0.0057 | 0\.8546$\pm$0.0071 |
> | 9    | 0\.8330$\pm$0.0093 | 0\.7949$\pm$0.0163 | 0\.8412$\pm$0.0084 | 0\.8403$\pm$0.0063 | 0\.8289$\pm$0.0456 | 0\.8124$\pm$0.0122 | 0\.8364$\pm$0.0086 |
> | Avg. | 0\.8550$\pm$0.0136 | 0\.8026$\pm$0.0118 | 0\.8591$\pm$0.0102 | 0\.8453$\pm$0.0101 | 0\.8265$\pm$0.0204 | 0\.8266$\pm$0.0131 | 0\.8564$\pm$0.0098 |
>
> The results show that the specific splitting generalizes well for most GML models. Previous works like OGB, despite the specific design of splitting, also use a single splitting (change of splitting will definitely influence its result). This is a common way to have unified settings for a benchmark. We consider this point and make our splitting scheme follow a specific distribution rather than totally random. Moreover, since we consider weighted accuracy across various methods for leaderboards, the metric has a tolerance for small variance.
> Nevertheless, we totally agree that aggregating results from more random splits will make the evaluation more general, and we will consider your advice to improve our work.
>
> 4. **Grammatical issues:** Thank you for pointing this out, we have fixed the errors and carefully revised the paper in the new version.
>
> We hope these responses address your concerns.

---

> > ### Comment · Reviewer_zT1n · 2021-09-29
> > **Addressed Concerns and a follow up**
> >
> > I would like to thank the authors for the response.
> >
> > ## Addressed Concerns
> >
> > * The authors have entirely addressed my concern for the longevity of this work by including a reasonable maintenance plan. I think the authors and I agree that given the nature of a benchmark of this sort, having a plan to update the work to remain current is vital. I believe the authors have proposed a reasonable maintenance plan.
> >
> > * The work still only incorporates one task, but I agree with the authors that they have provided a sufficiently robust and well documented framework that building off of it to extend to future tasks is defiantly possible, but it does require some task specific considerations. While I would still like to see more tasks, the work as-is is still useful in my opinion, so I would classify this concern as at least partially addressed. I hope to see an expansion of tasks and datasets in the future!
> >
> > ## Points where some uncertainty remains
> >
> >
> > I still have some concerns about the generalizability of this work in practice. I understand the authors claim that their split is more generalizable than using a single random split, and the novel splitting method proposed by the authors seems reasonable. However my concern was more about the fact that the results generated by this benchmark have only really been shown to hold for these datasets and for this splitting schema.
> >
> > For example, consider I'm trying to decided between two GML models ($\mathcal{M}_1$ and $\mathcal{M}_2$) based on their robustness to various attacks. To my understanding, depending on the splits used, the ranking of $\mathcal{M}_1$ and $\mathcal{M}_2$ may change. To standardize the evaluation of GML models, the authors propose a novel splitting schema so that within their benchmarks the rankings are consistent, which is a good idea so that you can compare attacks/defenses directly.
> >
> >  My concern is that given that their is no general way to split a given dataset (splitting is generally domain-dependant to my knowledge), is it the case that the ranking provided by this benchmark may not generalize to an actual use case? I'd imagine If I used this benchmark in this way, whatever model I deploy would have to use the same splits this model uses, which means I might need to change the current splits already used by my model (which may impact performance or lead to a model that is not usable in practice). Even if this benchmark wasn't meant to be used in this way (to decide which of a set of GML models to deploy based on robustness), would the splitting schema you present provide information about attack/defense effectiveness that does not hold when your splitting schema is not used? I understand that the authors designed this schema specifically to evaluate adversarial robustness, but I'm not yet convinced that this schema does not confound the effectiveness of various attacks/defenses in practice. I would appreciate the authors correcting me on any misunderstanding I may have about this, providing evidence that any conclusions derived from this schema do in fact hold with other splits used in production, or following up and adding the "split schema"  as a sort of variable for each given task.
> >
> > ## Summary
> > I still have some concerns about whether the results obtained from this benchmark can actually generalize to settings in practice. That said, given the framework of this model, at worst the authors could define a splitting schema per dataset/per task with relative ease and at best my concerns are overblown and the issue I percieve is not actually an issue in practice. While I would appreciate some clarification on that front, I think there is sufficient useful work here to accept the paper.

---

> > > ### Author Response · Authors · 2021-09-30
> > > **About the common challenge of generalizability in GML**
> > >
> > > We highly appreciate your deep thinkings about generalizability in GML, which inspires us a lot in how to further improve the current benchmark.
> > > The generalizability is actually a challenge faced by the entire GML community. As you mentioned, graphs are very domain-specific. We think this can be considered from two perspectives: *intra-domain* and *inter-domain*. Even for the famous OGB benchmark, we can observe that model architecture can’t generalize well across datasets, i.e. the best-performing model for one dataset is not guaranteed to perform well on another. This not only happens with inter-domain datasets, but also intra-domain ones (e.g. results on the small academic graph are different from the large one). A similar phenomenon can be found in GRB, there is no model that is always more robust across datasets, also discussed in Section 4 (Line 302-318).
> > >
> > > To better understand the generalizability in GML, we highly agree that the splitting scheme can be considered as a variable. And we propose two directions:
> > > 1. **For *intra-domain* generalizability:** The domain-specific splitting scheme of OGB was proposed with good motivation. There is room for improvement. The splitting is ad-hoc designed and fixed, e.g. splitting by time, it is hard to generate data with a similar distribution with a single graph. Thus to prove generalizability, we might need to have multiple graphs in the same domain for evaluation. Even though, graphs on the same domain can vary in properties, which leads to different conclusions. A potential solution is to use a customizable splitting scheme (fortunately this is naturally supported by GRB, the splitting scheme is a customizable function). Imagine for practical use, we are able to obtain multiple graphs with a similar distribution and properties (e.g. graphs obtained through many years). We can **calibrate the splitting scheme by a customized one** (can be very domain-specific), which makes the results more general for graphs on the same domain. We will include more datasets in the future, but for uncovered domains, customization of GRB still maintains its usefulness, e.g. one can build a customized dataset with a domain-specific splitting, and use ready-to-use models and attacks to test robustness in a practical case.
> > > 2. **For *inter-domain* generalizability:** We can also think if there exist general properties shared by inter-domain datasets. We propose one in our work, through the observation that nodes with lower degrees are generally more vulnerable for most graphs. In this case, it might be hard to directly find the best-performing model when there comes a new graph. Nevertheless, we can still have interesting findings, as some defense methods can generally increase robustness (model A + defense C is not necessarily better than model B + defense C, however, they might already be better than only using vanilla models). We think this is also helpful for the community, and we will investigate more splitting schemes in this direction.
> > >
> > > Overall, both perspectives are valuable, we will take the splitting scheme as an important factor to better understand generalizability in GML.
> > > Thanks very much for your very insightful thoughts!

---

### Decision · Program_Chairs · 2021-10-10

**Decision:**

Accept

**Comment:**

This is a close call, but overall, we feel like there is enough support for the paper from a majority of reviwers.